# Three Dimensional Change Detection Using Point Clouds: A Review

Abderrazzaq Kharroubi [1,*], Florent Poux [1], Zouhair Ballouch [1,2], Rafika Hajji [2] and Roland Billen [1]

1   UR SPHERES, Geomatics Unit, University of Liège, 4000 Liège, Belgium
2   College of Geomatic Sciences and Surveying Engineering,
    Hassan II Institute of Agronomy and Veterinary Medicine, Rabat 10101, Morocco
*   Correspondence: akharroubi@uliege.be; Tel.: +32-493443901

**Abstract:** Change detection is an important step for the characterization of object dynamics at the earth's surface. In multi-temporal point clouds, the main challenge is to detect true changes at different granularities in a scene subject to significant noise and occlusion. To better understand new research perspectives in this field, a deep review of recent advances in 3D change detection methods is needed. To this end, we present a comprehensive review of the state of the art of 3D change detection approaches, mainly those using 3D point clouds. We review standard methods and recent advances in the use of machine and deep learning for change detection. In addition, the paper presents a summary of 3D point cloud benchmark datasets from different sensors (aerial, mobile, and static), together with associated information. We also investigate representative evaluation metrics for this task. To finish, we present open questions and research perspectives. By reviewing the relevant papers in the field, we highlight the potential of bi- and multi-temporal point clouds for better monitoring analysis for various applications.

**Keywords:** 3D change detection; 3D point clouds; deep learning; machine learning; datasets



## 1. Introduction

The rapid development of 3D data acquisition is making the collection of massive point clouds faster than ever before, making them the future core topographic data for several applications [1]. However, in a dynamic world where everything is continually changing, data must be updated and used to detect and characterize the changes that have occurred. Change detection (CD) is defined as the process of recognizing the dynamics and changes in the earth's surface that occur between two or more epochs over the same geographic area [2].

CD has been widely studied in remote sensing applications, and many approaches based on two-dimensional (2D) images have already been proposed [3–11]. The availability of continuous large volumes of satellite images with short revisit times has created favorable conditions for the detection of long-term changes with improved temporal resolution. Similarly, 3D change detection (3D CD) is attracting more and more attention, due to the increasing availability of 3D data at different scales (district, city, region, and country). In real-world cases, change can range from rapid change (for example, in the context of autonomous driving [12,13]) to slow change (e.g., remote sensing applications [7,14,15]) depending on the studied phenomenon, its frequency, magnitude and velocity. We divide change into two main categories: 3D tracking of homologous parts of a surface to compute a displacement field (fast change) and computing the distance between two points clouds when the homologous parts cannot be defined (slow change). For each one, there are several methods to measure and track object dynamics. In 3D remote sensing applications, there has been increasing demand for 3D CD in the following fields: land use and land cover change detection [16], urban monitoring [17–19], forest changes [20–22], crisis monitoring [23],

3D geographic information updating [24–26], landslide and erosion monitoring [27–31], construction progress monitoring [32], and resources surveying [33,34].

When dealing with multi-temporal point clouds, we talk about a point cloud with reference to a specific time, "epoch". Due to the change in acquisition parameters from one epoch to another, it is not possible to make a direct comparison between two points clouds, because the sampling of points on the surface is not the same. Consequently, a point does not have its direct homologue on the second epoch, and, hence, no homology is possible between points; therefore, no direct calculation of point-to-point distance is possible. This aspect must be considered for 3D CD, either by preprocessing, if the density is different, or by using adapted methods (e.g., Cloud-To-Cloud) [35]. Change detection encompasses, in addition to displacement calculation and "from–to" change classification, other aspects, such as binary changes, multiclass changes, direction and magnitude of change, probability of change, temporal change trajectories (trends) and deformation analysis or abnormal comportment [14].

The 3D CD methods can be subdivided into Point-Based (PBCD), Object-Based (OBCD) [36–38], and Voxel-Based (VBCD) [39–41]. OBCD allows the detection of changes at an object level (segment or clusters that group a set of homogeneous points, or an instance that belongs to a known object class, like tree, car, building, etc.). The CD step is largely influenced by the quality of the detection and classification of objects. VBCD methods rely on the discretization of space into grids, octrees, or voxels (e.g., occupancy grids), which are mainly used in robotic and indoor mapping applications [42,43]. In this review, we focus mainly on the first category, based directly on points with referrals to other approaches, without going into excessive detail. Although PBCD is more popular, due to its simple algorithms and relatively better quantitative results, applying these methods to multimodal point clouds often produces incorrect results. CD of large and outdoor scene point clouds faces many challenges, including incomplete data, noise, artifacts caused by temporary or moving objects, and cross-source point clouds captured by different sensor types. To address these challenges, many studies have proposed different methods to use voxels and objects as basic units for change detection [38–40].

Another aspect that needs to be introduced is semantic segmentation. Segmentation of 3D point clouds is the process of classifying point clouds into several homogeneous regions, where points within a region have the same properties. Segmentation is challenging, due to the high redundancy, uneven sampling density, and lack of explicit structure of point cloud data [44]. With the development of deep learning for point cloud semantic segmentation [45–47], high-level point clouds with semantic information can be obtained at unprecedented scales. Inspired by this development, recent research tends to incorporate high-level semantic features into point clouds CD to solve the problems of classic CD pipelines, such as binary CD (change, no change) and missing information about change type. However, finding changes is only one aspect of the problem. A subsequent crucial task is to make sense of them. What causes changes? Why and when do changes occur? What impact could changes have on other objects? Therefore, the seemingly simple question of what can be considered a "change" in point clouds is not trivial.

In this paper, we offer a comprehensive review of change detection using 3D point clouds. We review the 3D CD methods used in remote sensing applications without integrating the 3D scene flow [48–50] or 3D object tracking [51,52] methods. We introduce distance-based methods and learning-based methods with a focus on deep learning-based ones. The main contributions of our paper are fourfold:

- Challenges related to the use of point clouds in CD and survey of 3D CD methods;
- Comprehensive review of the popular point clouds datasets used for 3D CD benchmarks;
- Detailed description of evaluation metrics used to quantify change detection performance;
- List of the remaining challenges and future research that will help to advance the development of CD using 3D point clouds.

We structure the rest of the review as follows. Section 2 reviews the challenges related to 3D CD and relevant works on point clouds CD. Section 3 describes a summary of existing 3D point cloud datasets and evaluation metrics. Section 4 proposes a list of the remaining challenges for future research. Section 5 closes the paper.

## 2. 3D Change Detection Using 3D Point Clouds

Our approach for compiling different works on 3D change detection using point clouds is based on previous reviews of the literature, as in [14,53]. This gave us a quick and easy way to situate the state of the art, as well as to understand the proposed taxonomies to categorize the existing approaches. To the best of our knowledge, based on the study of the literature, this is the first review dedicated to change detection using 3D point clouds to incorporate the latest machine and deep learning methods.

The articles reviewed in this paper were published between 2004 and 2022. Our contribution is mainly focused on approaches using 3D point cloud data. For the rest of this paper, we refer to change detection (CD) as the following setting: two points clouds or more, acquired at different times, covering the same area of interest, in which parts, objects, or surfaces move, change scale or color, distort, appear, or disappear between two different times. We assume, that point clouds are registered (aligned in the same frame reference), can have different sources (dense image matching, laser scanning, etc.) and be acquired from any platform (aerial or terrestrial, etc.). Unlike 3D CD, in 2D CD in remote sensing, a multitude of state of the art with most recent methods already exist [11,54,55]. Therefore, before reviewing the methods found in the literature, we discuss the issues related to the use of 3D point clouds, the challenges associated with them, and their advantages and disadvantages over 2D data.

### 2.1. Challenges and Specificities

2.1.1. Acquisition Challenges

The application of change detection on 3D point clouds presents many challenges. Indeed, changes in sensing conditions and unconstrained environments have a significant impact on the appearance of objects. Objects detected in different scenes or instances exhibit a range of variations. Even for the same scene, parameters such as scan timing, location, weather condition, sensor type, sensing distance, background, etc., produce significant variations for intra- and inter-class changes in 3D point clouds. We subdivide the major challenges related to the use of point clouds for CD into two key components:

Scan-related artifacts. All sensors are noisy, and the scanning system itself presents several artifacts which can be significant for applications of CD. One major source of artifacts on a scan is regions with non-diffuse reflection properties that refer to areas where pulses have been transmitted, but due to pulse absorption or specularity, not enough energy (if any) is returned to trigger a distance measurement (e.g., windows, water, and low reflectance surfaces). When data from another scan exists in these "hole" regions, it is considered a change while it is not. There is also, occasionally, noise caused by the presence of particles in the air when scanning, as well as unwanted points caused by the reflection of the laser pulse on a surface, such as a lake or a river. Hence, it is important to conduct a pre-processing step to clean up these artifacts before performing the change detection [56].

Occlusion. Due to occlusions, point cloud data are often incomplete. The occlusion of objects, or parts of objects, offers another variety of "interference", with the probable consequence of false CD. Points that appear in one scan but not in the other are considered potential candidates for natural change. To handle occlusion effects in point clouds, Ref. [57] proposed a new strategy for detecting "changed", "unchanged", and "unknown" buildings, where the latter class is applied to places where, due to lack of data in at least one of the epochs, it is not possible to reliably detect structural changes. Other research papers address this problem by using deep learning to fill in the occluded parts [58–64].

### 2.1.2. Three Dimensional Point Clouds Specificities

For any type of data to be processed, either in 2D or 3D, there are specific characteristics to consider. So, in addition to the point clouds challenges associated with acquisition, storage and manipulation phase, there are other challenges associated with the processing stage. We detail four challenges specific to point clouds that must be considered for each application, or algorithm, and, thus, be taken into account for CD.

Irregularity. Due to several acquisition parameters (e.g., the distance of the point from the sensor), point cloud data are irregular. This means that points are not homogeneously scattered in different regions of an object or scene, so some regions may have dense regions and others sparse ones, resulting in a changing density inside the same point cloud, as we can see in Figure 1a. This irregularity can be reduced by subsampling techniques using octree or spatial distance, but cannot be eliminated [41].

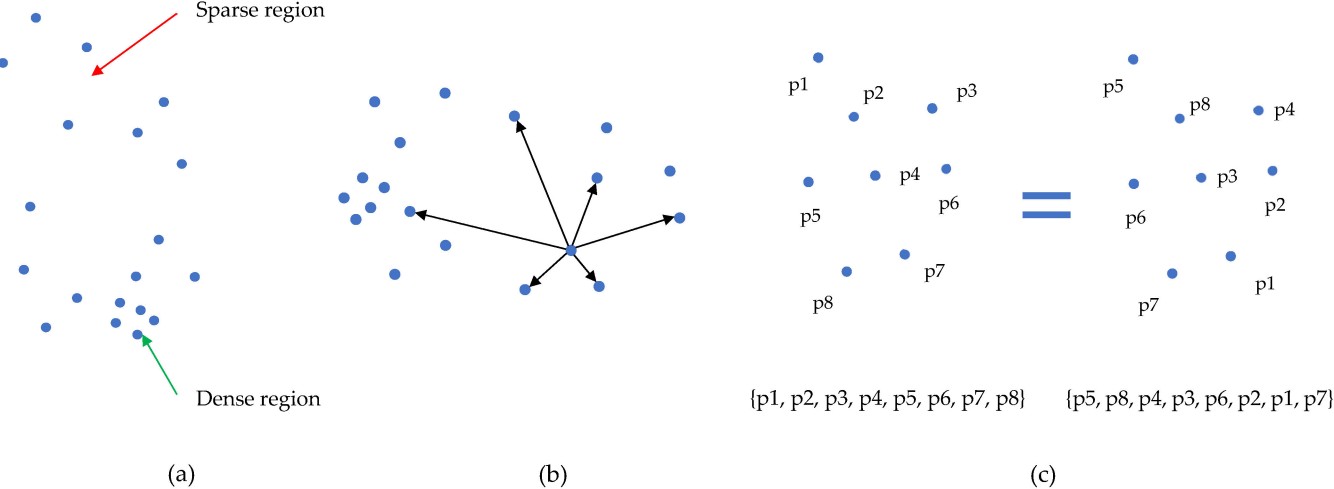

**Figure 1.** Challenges related to 3D point cloud processing. (**a**) Irregular: with dense and sparse regions. (**b**) Unstructured/no grid: each point is independent and the distance between adjacent points is not fixed. (**c**) Unordered: as a set, point clouds are permutation invariant.

Unstructured. Unlike images, point clouds in their raw form are not placed on a regular grid. Each point is scanned independently, and its distance to adjacent points is not always fixed, as shown in Figure 1b, which makes their spatial structuration complex. In analogy to images where pixels are represented on a two-dimensional grid, and the space between two adjacent pixels is always fixed, the point cloud is sometimes transformed into voxels to facilitate processing [39,65].

Unorderedness. A point cloud of an object or a scene is a set of points derived from the object's surface (represented by XYZ coordinates and other attributes). They are usually stored as a list in a file. As a set, the order in which the points are stored does not change the represented scene, so we say it is a permutation or order invariant. The unordered nature of point sets is illustrated in Figure 1c.

Rigid transformation. There are various rigid transformations in point clouds, such as 3D rotations, scale, and translations. However, these transformations should not affect the results and performance when using processing algorithms and especially deep neural networks.

To date, change detection in remote sensing has been primarily image-based, typically using object–background separation, simple subtraction between images, and, more recently, learning-based methods. These approaches have the disadvantage of imposing rigid constraints, such as static camera, the need to have the same viewpoint of the satellite, the need for recognizable landmarks, and sensitivity to shadowing and local illumination problems. The advantages and disadvantages of 2D data for CD have been known for

a long time, while 3D CD work has emerged recently, mainly due to the availability of multi-temporal point clouds and the development of learning-based algorithms (Table 1).

**Table 1.** Summary of main differences between 2D and 3D CD specificities [14].

| | 2D CD | 3D CD |
|---|---|---|
| Data source | Optical images, multi spectral images, RADAR images [66], Digital Surface, Terrain, and Canopy Models, and 2D Vector data. | Point clouds, InSAR (Interferometric SAR), Digital surface model, stereo and multi-view images, 3D models, building information models, and RGB-D images. |
| Advantages | Well-investigated [7,67–69], available datasets [70–75], available implementation [68,74,76] | Height component, Robust to illumination differences, Free of perspective effect, and provide volumetric differences. |
| Disadvantages | Strongly affected by illumination and atmospheric conditions. Limited by viewpoint and perspective distortions. | Unreliable 3D information may result in artifacts. Limited data availability. Expensive processing. |

Despite the issues associated with using point clouds as a type of 3D data, they have several advantages which motivate their use for change detection. The main advantages of using 3D data over 2D data for change detection are summarized as follows: (1) Insensitive to illumination difference. As already stated, point clouds refer to spatial measurements of 3D objects; therefore, the comparison of the geometry of multi-temporal data is independent of illumination conditions; (2) Insensitive to perspective distortions. Using point clouds, geometry comparison can be performed in a real three-dimensional space, or any projected 2D space (subspace of the 3D space). In this case, they are not influenced by the point of view, as for 2D images where this effect is very noticeable; (3) Volume information. Change detection in 3D provides information on volumetric changes which paves the way for more applications, such as volumetric loss of forests, precise monitoring of construction progress, etc.

*2.2. Data Preprocessing*

In remote sensing, image preprocessing is an essential first step, it includes geometric rectification and images registration, radiometric normalization, cloud and cloud shadow detection, atmospheric and topographic correction [77,78]. For 3D point clouds, preprocessing is an essential step before applying different change detection algorithms. The purpose is to minimize changes due to characteristics we are not interested in, and to identify changes we are interested in. So, we make the data at different epochs comparable. It includes removal of outliers, filtering, registration, and rasterization of DSMs (Digital Surface Models), DEMs (Digital Elevation Models) and nDSMs (normalized DSMs). We detail each step as follows:

Removal of outliers. The first step in preprocessing is the removal of outliers (unwanted points). They can be moving objects (a person, a car, etc.), vegetation, acquisition artifacts, noise, or points outside the area of interest. By removing these undesirable points, errors are reduced in the subsequent steps, such as normal calculation, registration, rasterization and change detection. The removal can be done manually by the operator or automatically, based on adapted algorithms [79–81]. In [28], the outlier removal algorithm calculates, for each point, the distance to all its neighbors and removes points having distances outside the point cloud's global mean and standard deviations.

Filtering. The purpose of filtering is to separate ground points from non-ground points. Several methods exist for this purpose. Ref. [82] classifies them into three categories, namely, mathematical morphology-based filters [83–86], surface-based filters [87–89], and slope-based filters [82,90,91]. A further category can be considered which is segmentation-based filters [92–95].

Registration. Multi-temporal point clouds need to be aligned before performing changes analysis [28]. They do not need to be acquired from the same view, unlike images,

but must be georeferenced in the same coordinate system. For a point cloud acquired from a single station/viewpoint, if the sensor moves between epochs, if different sensors are used at different times or if at least parts of the surfaces deform between measurements, the same points are not measured or cannot be easily identified in the point clouds. Several algorithms exist for this task, comprehensive reviews of which can be found in [96–98].

Rasterization. For methods based on DSM, DEM or nDSM (see Figure 2 for the difference) for 3D CD or multimodal methods which take different types of data as input, rasterization allows derivation of a DSM from the point cloud. First, the point cloud is structured into a grid (e.g., cell size 1 m). Then, the lowest point of each cell of the grid is selected for triangulation as described in [99]. A similar method can be used for the rasterization of DEMs, but only the ground points are used in the triangulation. The nDSMs are derived through subtraction of the appropriate DEM from the DSM, as specified in the next formula:

$$nDSM = DSM - DEM \qquad (1)$$

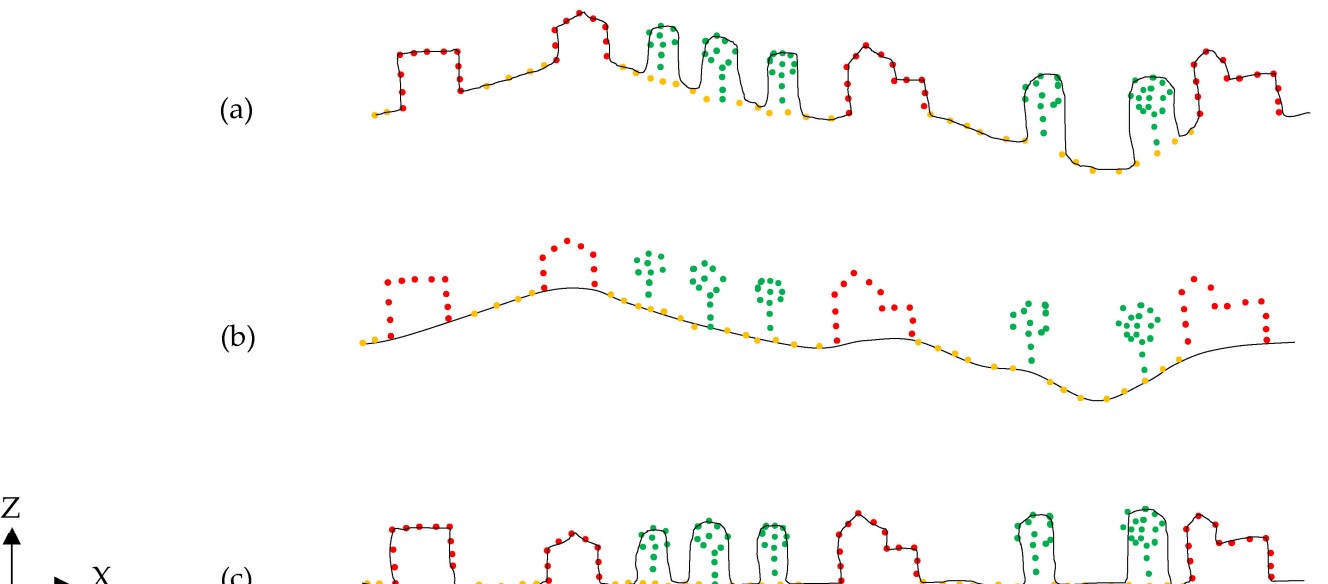

**Figure 2.** Schematic diagram to show the difference between: (**a**) DSM (Digital Surface Model), (**b**) DEM (Digital Elevation Model), and (**c**) The nDSM (normalized Digital Surface Model).

### 2.3. Three Dimensional Change Detection Methods

It is important to note that change detection techniques can be applied to a variety of input data (3D point clouds, meshes, 2D images or a combination of both). In our paper, we focus on 3D point clouds. Based on our review, we find that the categorization of these techniques can be done in different ways based on several criteria:

- Change Unit. This taxonomy depends on the basic unit used in the CD process, such as methods based on points, voxels, objects and rays;
- Order of classification and change detection. In this categorization, difference exists between methods that proceed to change detection and then classification (pre-classification methods), those that proceed to classification first (post-classification methods), and those that integrate the two steps into one (integrated);
- Used technique. Methods are classified here based on the used technique whether it is based on distance or learning, etc;
- Target. This depends on the application domain: urban, forestry, maritime, etc.

Many different point cloud-based change detection techniques have been proposed in different contexts of remote sensing, self-driving vehicles or robotic applications. In general, many methods use point clouds to compare 3D representations of the environment

in different states. Below (in Table 2), we summarize the methods found in the literature and classify them according to the used input data (LiDAR, images which can be remapped into orthophotos or the originals ones, and maps), the order of classification and change detection (integrated, pre- and post-classification change detection) and the target context (building, tree, vegetation, etc.). In addition to these criteria, we indicate the methods that transform the point clouds into a DSM (Digital Surface Model) to work on it. Most of the works tend to use point clouds more often and integrate classification and change detection in one step to overcome the limitations related to the other types. We can also see that most of the studies concern the urban environment and, more specifically, buildings.

**Table 2.** Overview of 3D CD methods by input data, approach, and change detection class.

| Authors | Year | Input Data | | | Change Detection Approach | Change Detection Class |
|---|---|---|---|---|---|---|
| | | LiDAR | Image | Maps | | |
| Matikainen et al. [100] | 2004 | X | Ortho | X | Post-classification | Building |
| Vu et al. [101] | 2004 | X | | | Pre-classification DSM-based | Building |
| Vosselman et al. [102] | 2004 | X | | X | Post-classification | Building |
| Choi et al. [103] | 2009 | X | | | Post-classification | Ground, vegetation, Building |
| Matikainen et al. [104] | 2010 | X | Ortho | X | Post-classification | Building |
| Stal et al. [105] | 2013 | X | Ortho | | Post-classification | Building |
| Malpica et al. [106] | 2013 | X | Original | | Post-classification | Building |
| Teo et al. [107] | 2013 | X | | | Post-classification DSM-based | Building |
| Pang et al. [56] | 2014 | X | | | Pre-classification DSM-based | Building |
| Zhang et al. [108] | 2014 | X | | | Pre-classification | Ground |
| Tang et al. [109] | 2015 | X | | X | Post-classification | Building |
| Awrangjeb et al. [26] | 2015 | X | | X | Post-classification | Building |
| Xu et al. [57,110] | 2013, 2015 | X | | | Post-classification | Building |
| Xu et al. [57,111] | 2015 | X | | | Pre-classification | Building, tree |
| Du et al. [112] | 2016 | X | Original | | Pre-classification | Building |
| Matikainen et al. [113] | 2016 | X | Ortho | X | Post-classification | Building |
| Matikainen et al. [114] | 2017 | X | Ortho | X | Post-classification | Building, roads |
| Kaiguang et al. [115] | 2018 | X | | | Post-classification | Forest |
| Marinelli et al. [116] | 2018 | X | | | Post-classification | Forest |
| Zhang et al. [117] | 2019 | X | Ortho | | Integrated | Building |
| Zhang et al. [118] | 2019 | X | Ortho | | Integrated | Building |
| Yrttimaa et al. [22] | 2020 | X | | | Post-classification | Forest |
| Fekete et al. [119] | 2021 | X | | | Post-classification DSM-based | Tree |
| Huang et al. [120] | 2021 | X | Original | | Post-classification | Building |
| Ku et al. [121] | 2021 | X | | | Integrated | Building, street, tree |
| Iris et al. [122] | 2021 | X | | | Integrated | Building |
| Tran et al. [123] | 2021 | X | | | Integrated | Ground, vegetation, building |
| Zhang [124] | 2022 | X | | | Integrated | Building |
| Dai et al. [36] | 2022 | X | | | Integrated | Building |

The new categorization we propose is based on the algorithms used for change detection, regardless of the basic unit, the context studied or the order of classification and change detection. It classifies change detection methods into three main types that can, in turn, incorporate subtypes. The first type includes standard methods (also called distance-based methods), the second type includes machine learning-based methods that use handcrafted features, and the last type includes the deep learning methods that extract more abstract features without user specification (see Figure 3).

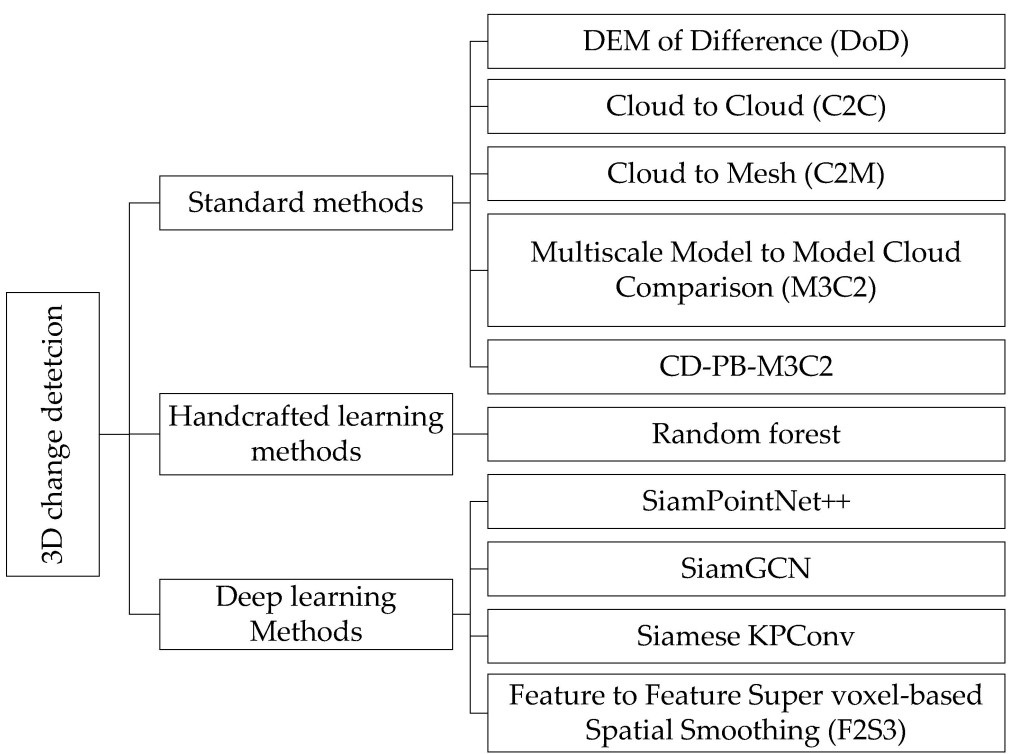

**Figure 3.** The taxonomy of 3D CD approaches with the methods we will detail in the next sections.

2.3.1. Standard Methods

This category of 3D CD includes point clouds comparison methods based on the calculation of the distance between two points. They establish displacement based on proximity in Euclidean space. There are approaches based on the difference between digital elevation models (DEMs), the distance between point clouds using the nearest distance, and multi-scale point clouds comparison. We detail each of these as follows:

DEM of difference (DoD). This has a simple concept and easy implementation; it quantifies the surface change based on the DEM derived from a 3D point cloud. One of the most common methods in this category is the difference between DEMs that estimate elevation change on a cell-by-cell basis where the change is derived along a single, predefined direction (Z-axis) [125–127]. This simplicity reveals a limitation in complex contexts, such as overhangs and near vertical slopes, where the vertical difference is not sufficient.

Cloud-to-cloud comparison (C2C). Point cloud comparison is the simplest and most effective method for deformation and CD [35]. The change is detected by calculating the Euclidean distance between individual points in a reference point cloud (epoch A) and the respective nearest neighbor point (NN) in the target point cloud (epoch B). We illustrate this process in Figure 4a. Its simplicity is its major advantage, since it does not require the calculation of normal, and, therefore, no local modeling of the surface is needed (e.g., triangulation or plane adjustment).

Cloud-to-Mesh comparison (C2M). Like the C2C method, the C2M method computes displacements using the nearest Euclidean distance between each point in a source point cloud and the nearest facet (or to an edge, if the orthogonal projection of the point does not fall on any facet) in the target mesh of the triangulated point cloud [128,129]. The principle of this method is illustrated in Figure 4b. Its main limitation is its need to mesh the target point cloud. This can generate triangular surfaces with holes and artifacts, which leads to false displacement and change detection.

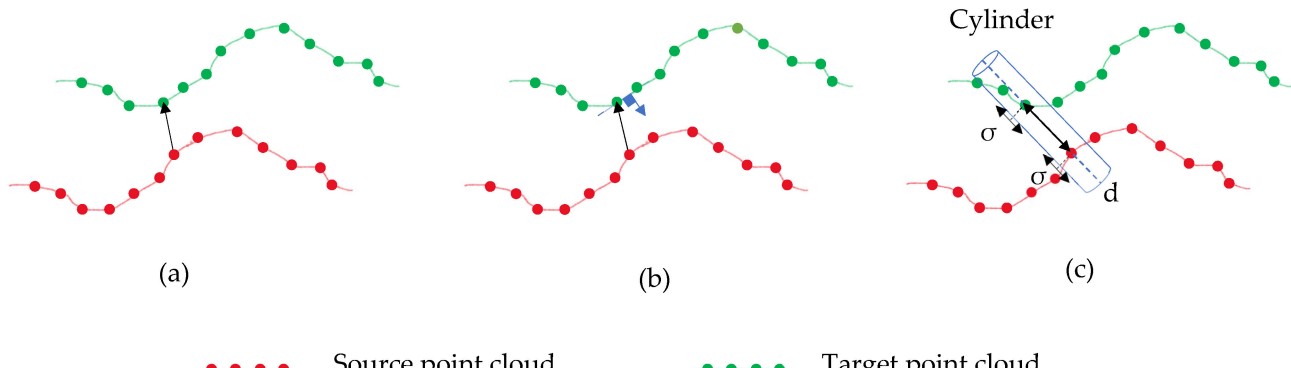

**Figure 4.** Existing approaches of point-to-point distance calculation. (**a**) Cloud to Cloud. (**b**) Cloud to Mesh. (**c**) Multiscale Model to Model Cloud Comparison.

Multiscale Model to Model Cloud Comparison (M3C2). This method estimates the displacement based on the points of interest found in each point cloud (e.g., sub-sampled points of the point cloud or even the entire source point cloud) [130]. It has two main steps: (1) Surface normal estimation and orientation in 3D space at a scale consistent with the locale surface roughness; (2) Measurement of the mean surface change along the normal direction with the explicit calculation of local confidence interval (σ). Its main advantage is that it works directly on point clouds without needing to mesh or grid them. It estimates a confidence interval for each distance measurement depending on the roughness and registration error of the point clouds (see Figure 4c). However, the M3C2 performs less well when the changes occur in different directions than the direction of change computation, and also when the level of detection (see Section 3.2) exceeds the magnitude change.

There are two problems associated with this type of methods, which relies on the point-to-point distance calculation for CD. The first issue is related to density, which is variable between two epochs and within the same epoch. This is affected not only by the distance of the point from the acquisition sensor but also by the change in the type of sensor and the acquisition mode. The second problem is related to the inefficiency of these methods to deal with occluded areas in point clouds, because they do not consider free spaces. To counter these two problems, there are ray-based methods. These methods require that the sensor positions are known for each instant to recreate a bundle of rays representing the pulse path and the measured point [53]. However, this type of method is highly point-of-view dependent and cannot be generalized to data without information about the sensor positions.

A further challenge in 3D CD is the quantification of small changes with low uncertainty. A recent paper proposed an improved version of M3C2, called Correspondence-Driven Plane-Based M3C2 (CD-PB M3C2) [131]. Based on two points clouds at epoch A and epoch B, this method uses a three steps workflow (Figure 5a). The first is to extract the planar surface using a region-growing segmentation (Figure 5b). The second is the plane's correspondence search through a binary random forest classifier (RF) (Figure 5c). The third is the quantification of change and uncertainty (level of detection) through the calculation of M3C2 distance between each plane and its corresponding one (Figure 5d). This approach, based on plane correspondence, gives results seven-fold better than the M3C2, in terms of uncertainty associated with topographic change, and shows high performance for quantifying small-magnitude (less than 0.1 m) changes. The other main advantage of the CD-PB-M3C2 is that, by using the matching planes, it is not necessary to determine the direction of change a priori, as the feature similarities are used regardless of the absolute position of the planes. Nevertheless, the use of planes constitutes its own disadvantage because it does not allow high-level recognition of objects other than planes.

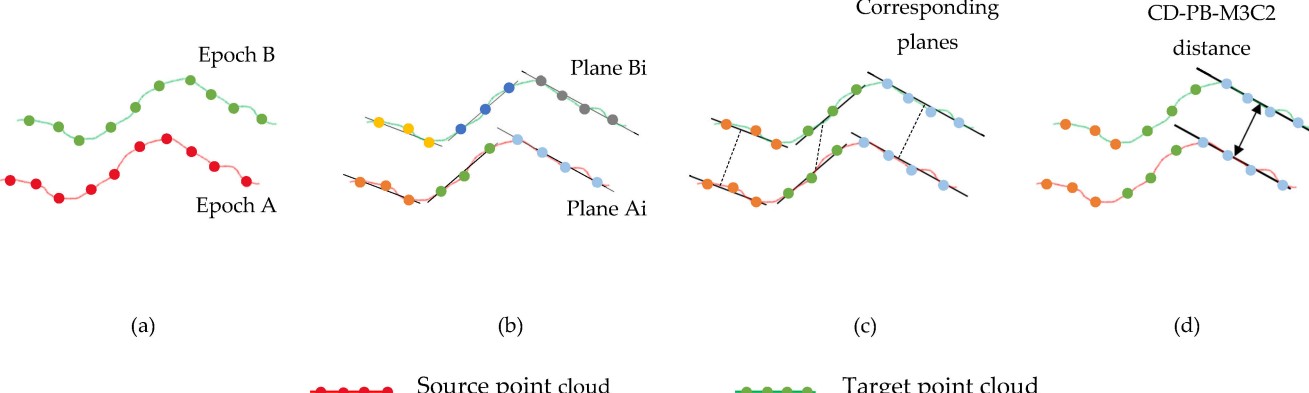

**Figure 5.** Correspondence-driven plane-based M3C2 as described in [131]. (**a**) Input point clouds. (**b**) Extraction of planar surfaces. (**c**) Plane correspondence search. (**d**) Quantification of change and uncertainty.

To summarize, the problem with traditional methods for CD and displacement analysis is establishing the correspondences between point clouds in the Euclidian space. However, these correspondences just represent the distance between the surfaces, not the actual displacement of points on the surfaces. Therefore, previous methods (C2C, C2M, M3C2 and CD-PB-M3C2) struggle to estimate the correct and significant change in a specific case, mainly in the case of parallel motion to the surface. To resolve this problem, several methods have been proposed in the literature, such as the fusion of point clouds and RGB (Red, Green, Blue) images, as in [132]. The latter estimates 3D change and displacement vectors using the corresponding points obtained, based on 2D RGB-Depth image in features space, as illustrated in Figure 6. The matching is not distance-based but uses the correspondence of the extracted features around each point in a radius *r* to find the corresponding point in the target point cloud. With the same consideration, recent machine and deep learning methods propose to establish correspondences in the feature rather than the Euclidean space only.

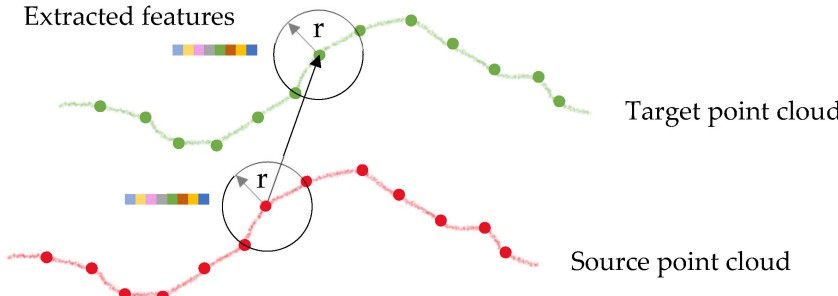

**Figure 6.** Correspondence between points at different epochs in features space.

### 2.3.2. Machine Learning with Handcrafted Features

In this section, we show works using handcrafted features in machine learning algorithms. These recent approaches for CD using 3D point clouds integrate the classification and change detection at the same time using handcrafted features [19,123]. The general idea of these methods is to extract inter- and intra-epoch features and then classify them using a learning model to obtain change classification results, as illustrated in Figure 7.

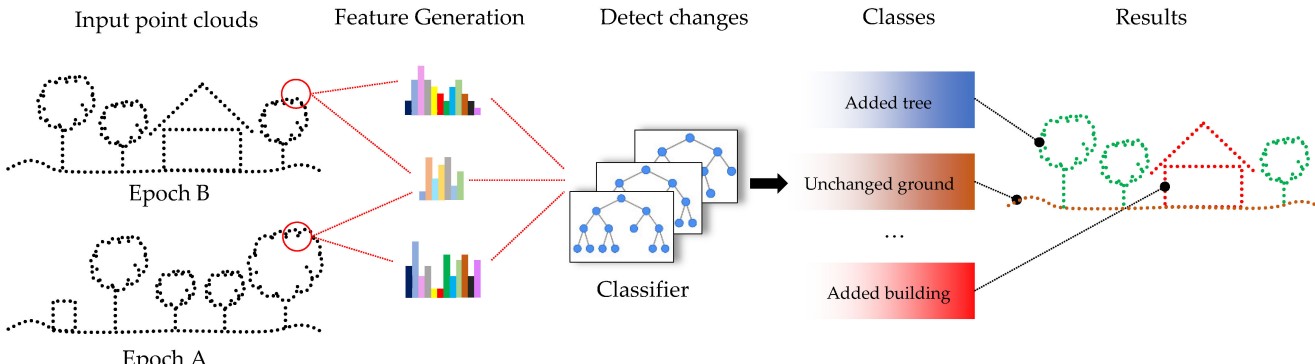

**Figure 7.** General 3D CD framework using handcrafted features.

For more details, we present the workflow proposed by [123]. Point clouds at two different epochs (A and B) are merged to extract four types of features: features describing the point distribution (e.g., planarity, verticality, linearity and omnivariance), feature height above the terrain, features specific to the multi-target capability of laser scanning (e.g., return number and number of returns) and features combining point clouds from both epochs to identify change. The proposed methods take as input two cleaned and georeferenced point clouds in the same coordinate system (no registration needed). Then, the point clouds are merged to generate the four specified features, as shown in Figure 8, and, finally, provide change detection classification type using random forest classifier.

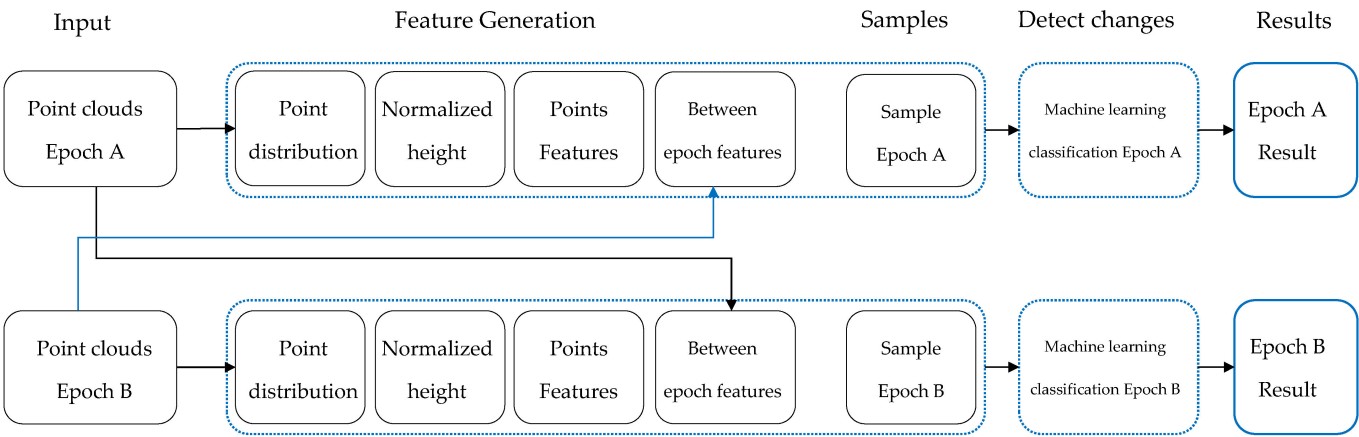

**Figure 8.** Integrated CD and classification of point clouds using handcrafted feature as in [123].

Despite its advantages of fusing classification and CD in a single step, as well as being able to detect change in multiple classes simultaneously (e.g., tree, building, soil, etc.), machine learning methods depend heavily on the initial training data. This problem can be partially solved by using unsupervised machine learning methods.

### 2.3.3. Deep Learning Methods

To overcome the problem of change detection based only on distance correspondence in a Euclidean space, deep learning methods propose to create more abstract features without the need for user specification. In [133,134], the authors proposed a new deep learning framework for point clouds displacement and change analysis, called Feature to Feature Super voxel-based Spatial Smoothing (F2S3). It is divided into two mains step: (1) Estimation of an initial 3D displacement vector field by determination of point-to-point correspondence in the feature space, and (2) Filtering and smoothing of the initial 3D vector field pipeline (see Figure 9). The proposed concept of F2S3 is not to rely on proximity in Euclidean space (distance) but to create a correspondence between points at different

times based on proximity in feature space. This proximity is covered by local feature descriptors, which describe the geometric information of the local neighborhoods of the point of interest (e.g., a sphere of radius *r*, as specified in Figure 6). Spectral and radiometric features (e.g., color, intensity, or multispectral bands) can be used in addition to geometric features, but these are usually neglected and not considered in the local feature descriptors, because of constraints related to changing acquisition conditions, environment, sensors, etc. Thus, by establishing the corresponding points within the feature space, F2S3 is sensitive to displacements along the surface. It was demonstrated in this work that F2S3 outperformed the standard methods (C2C, C2M, M3C2) on real-world geo-monitoring datasets when the hyper-parameters were chosen appropriately [133,134].

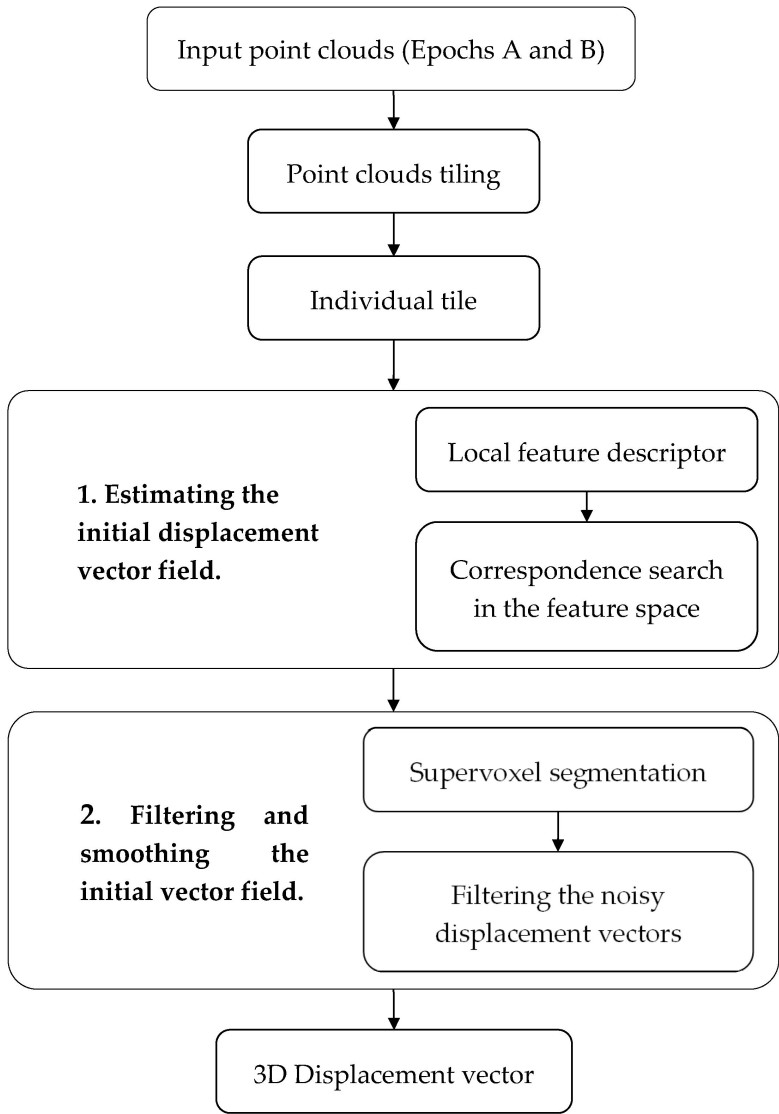

**Figure 9.** Workflow of the modified F2S3, as described in [30].

However, the available current implementation of F2S3 is computationally very complex, not fully automated, and requires in-depth knowledge of the algorithm and the deformation process to choose the right hyper-parameters. To solve such problems, Ref. [30] proposes a further step and integrates the F2S3 workflow into a fully automatic pipeline (Figure 9). They propose a tiling procedure to create smaller point tiles to facilitate their processing in an efficient and parallel way (to process large point clouds). They also propose to replace the hyper-parameters that required in-depth knowledge of the user with values derived directly from the input point clouds. To overcome the problem of processing time

and memory complexity, they propose using more efficient local feature descriptors, as proposed in [135]. These learning-based methods have been shown to outperform all the traditional methods already mentioned in Section 2.3.1.

In another work, Ref. [121] proposed a Siamese Graph Convolutional Network (SiamGCN) for 3D point clouds CD. The edge convolution (EdgeConv) operator is adopted to extract representative features from point clouds (Figure 10). Then, a Siamese architecture, [136], based on the graph convolutional networks, is proposed to identify the change type of any two input point clouds (A and B) from two different epochs. The source code of their approach is publicly available at https://github.com/kutao207/SiamGCN (last accessed on 28 August 2022). The authors also evaluated three algorithms, including one handcrafted and two learning-based methods, on the 3D CD dataset (more details in Section 3.2). The first one is point clouds change detection with hierarchical histograms (named PoChaDeHH). The second one is 3D point cloud CD for street scenes (named HGI-CD). The third is the SiamGCN (see Figure 10). Although the handcrafted algorithm can achieve relatively balanced results on the overall and per-class accuracy and mean intersection over union (mIoU), it was obvious that learning-based methods achieved overwhelming performance.

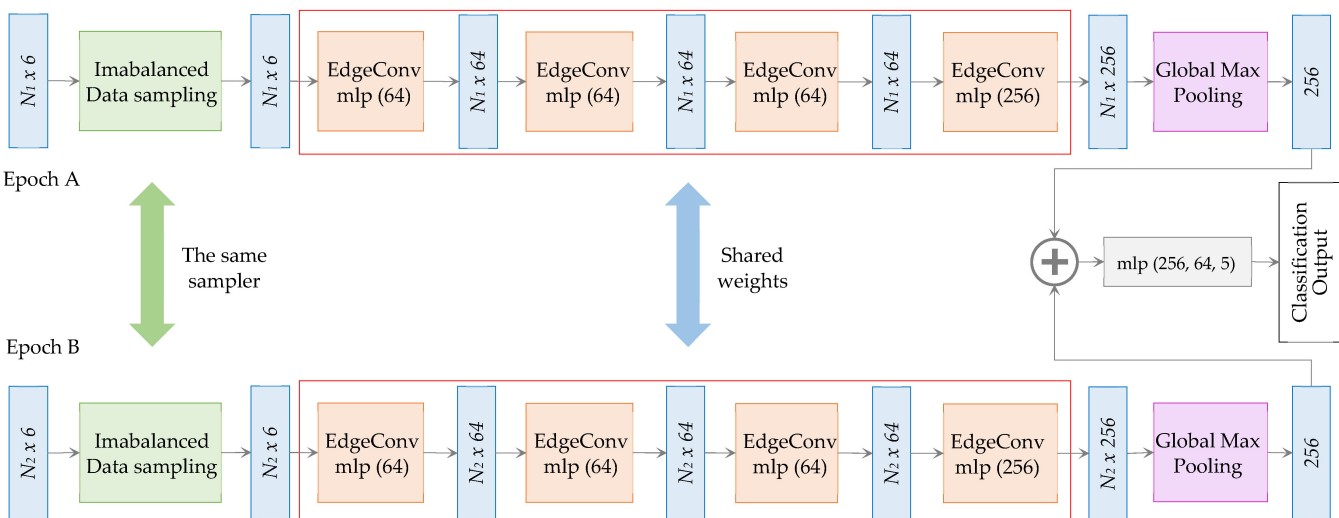

**Figure 10.** SiamGCN network architecture, as described in [121].

In [124], the authors proposed a method to detect building changes between LiDAR (Light Detection And Ranging) and photogrammetric point clouds. With consideration of the fact that semantic segmentation and CD are correlated, they suggested the Siam-PointNet++ model to combine the two tasks in one framework (see Figure 11). The method outputs a pointwise joint label for each point. If a point is unchanged, it is assigned a semantic label (e.g., building) and if a point is changed, it is assigned a change label (new building). The semantic and change information is included in the joint labels with minimum information redundancy.

The combined Siamese network learns both intra-epoch and inter-epoch features. Intra-epoch features are extracted at multiple scales (sphere radius *r*) to embed the local and global information. Inter-epoch features are extracted by Conjugated Ball Sampling (CBS) and concatenated to make change inferences (Figure 11). For the decoder layers, the DIM feature vectors are interpolated to the raw LiDAR points locations, instead of the raw Dense Image Matching (DIM) points locations. This ensures that the DIM features are calculated at the same centroids as the LiDAR data. Only feature vectors extracted at the same centroids can be compared. The authors point out that even if there is no DIM point in the conjugate ball of a LiDAR point, a pseudo feature map is calculated at the

same centroid to "inform" the model that the neighborhood of the ball in the DIM data is empty [124].

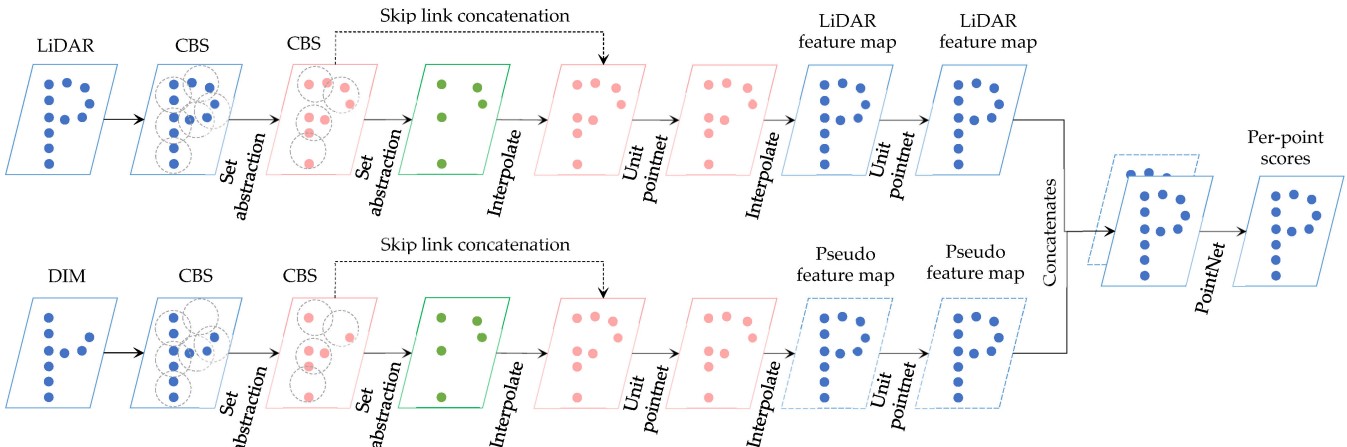

**Figure 11.** SiamPointNet++ network architecture [124].

Experiments conducted in a study area in Rotterdam, Netherlands, indicated that the network was effective in learning multi-task features. It was invariant to the permutation and noise of inputs and robust to the data difference between LiDAR and dense image matching data. Compared with a sophisticated object-based method and supervised change detection, this method requires much fewer hyper-parameters and less human intervention, but achieves superior performance, as stated by the authors.

The same author proposed in [117] a Convolutional Neural Networks (CNN) architecture for multimodal CD. Change in buildings was detected between a digital surface model (DSM) derived from point clouds, and a dense image matching point clouds using feed-forward CNN, which showed high performance for using multimodal data.

Using the same analogy of the Siamese architecture for 2D CD, Ref. [122] proposed to extend the Siamese to 3D point clouds. They proposed embedding the KPConv [137] architecture used for semantic segmentation into a deep Siamese network where both point clouds would pass through the same encoder with shared weights. Similar to the usual encoder–decoder architecture with skip connections, at each scale of the decoding part, they concatenated the difference of extracted features associated with the corresponding encoding scale, as shown in Figure 12. The authors compared the Siamese KPConv results to machine learning hand-crafted methods presented previously in [123]. The comparison results showed that the deep learning method outperformed the machine learning one for all evaluation metrics. An improvement of about 27 points of IoU (Intersection over Union) over classes of change was observed.

All conducted works in this section have shown that deep learning methods largely outperform traditional methods, either in terms of change classes number or in terms of evaluation metrics. Their main advantages lie in the ability to understand structured objects at a global scale, thus, leading to correct classification of hidden parts, and so resolving the occlusion part in point clouds. However, the problem of occlusions and variability of density and distribution of points between the different epochs is not completely solved. As can be noticed, learning-based methods largely depend on the availability of training data to achieve high CD quality. In the following section, we present multi-temporal point clouds benchmarks which are accessible in Open Access, as well as the metrics used in the literature for performance comparison.

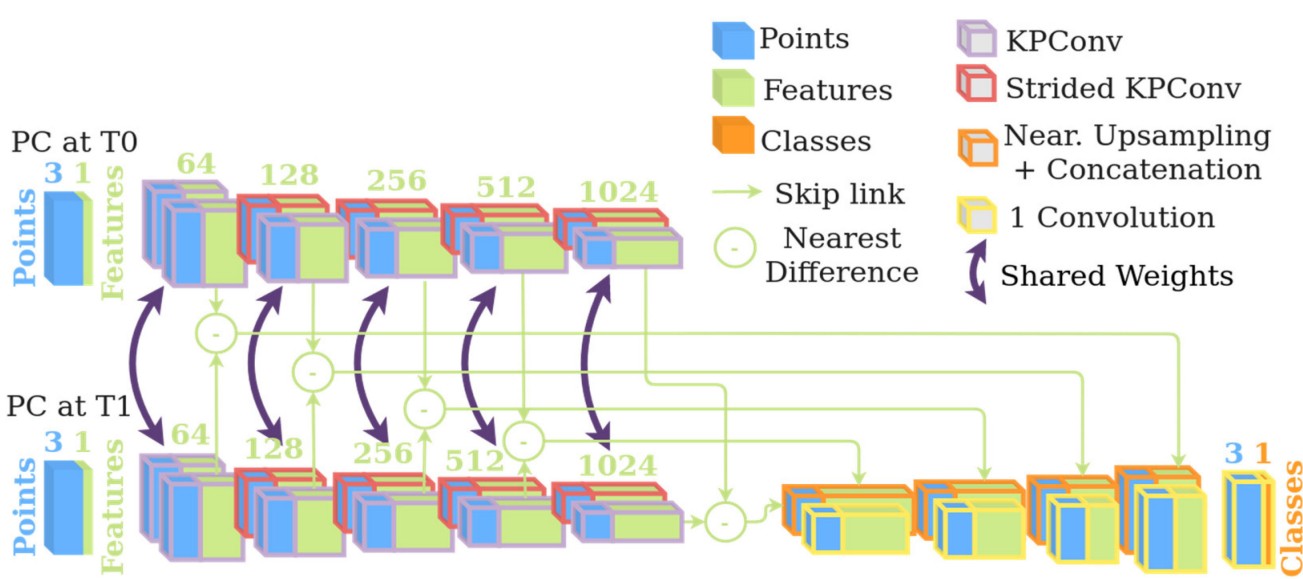

**Figure 12.** Siamese KPConv network architecture. Source: Image adopted from [122].

### 3. Benchmarks

A variety of methods are proposed to detect changes in the literature, but the choice of the right one is not obvious. Therefore, it is essential to adopt appropriate metrics to use for performance evaluation in the same dataset. In this section, we list publicly available benchmark point clouds datasets and provide some standard evaluation metrics used to compare the performance of the algorithms, for application in future research.

#### 3.1. Datasets for 3D Change Detection

This section presents different types of commonly used point clouds datasets for 3D CD. We classify the existing datasets related to our topic into two main types: unannotated datasets, which do not contain semantics information about objects classes or change type, and annotated datasets. The latter is the most useful because they pave the way for the rapid development of CD applications and exploitation of 3D data using machine learning methods and deep learning networks.

OpenTopography. This is a United States National Science portal that facilitates access to high-resolution data and related tools and resources [138,139]. It provides open access to LiDAR and photogrammetry point clouds and photos with on-demand processing tools to generate derived products. The one we are interested in is the change map created from the available multi-temporal point clouds. One can download point clouds acquired at different locations in high resolution. It also offers the possibility to use two services on demand: the first one is vertical differencing, which aims to measure landscape change by differencing DEMs (Digital Elevation Models) to see the topographic change from processes, including urban growth, flooding, landslides, wildfires, and earthquakes [140]. The second one is 3D differencing, which aims to detect the horizontal and vertical change when the landscape shifts during earthquakes and landslides [140]. This dataset is not annotated into change classes, and it is accessible at: https://opentopography.org/ (last accessed on 28 August 2022).

AHN1, AHN2, AHN3, AHN4. The Actueel Hoogtebestand Nederland (AHN) is the digital height map for the whole of the Netherlands. What is interesting about this is that there is repeat LiDAR acquisition AHN1, AHN2, AHN3 and a recent AHN4 [141]. AHN1 (1997–2004) was initiated by Waterboards, Ministry of Infrastructure and Water Management and Provinces to manage the water systems and water security, then AHN2 (2007–2012) and AHN3 (2014–2019). There is an upcoming version, AHN4 (2020–2022), which is denser than ever before. These datasets already contain a pre-classification,

but are not annotated in type of change, link https://www.ahn.nl/ (last accessed on 28 August 2022).

Abenberg—ALS test dataset. This dataset contains the point clouds acquisition of Adenberg, Germany (49.2416° N, 10.9636° E), using aerial laser scanner (ALS) RIEGL LMS-Q560 (version 2006). The first was acquired on 18 April 2008, by four ALS strips in a cross pattern, resulting in an accumulated point cloud which includes 5,400,000 points (Figure 13) with an average point density of 16 pts/m$^2$. The second was acquired on 31 August 2009, using the same sensors and a similar setting as specified by [142], resulting in a point cloud of 6,200,000 points with an average point density of 21 pts/m$^2$. In addition to the coordinates of 3D points (XYZ), the data sets contain the local normal directions, sensor positions, and results of pre-classification (ground, vegetation, and building). These multi-temporals multi-view point clouds data are well suited for the development and evaluation of 3D CD methods in urban areas, and the investigation of other applications of ALS data, e.g., city modeling and city model updating.

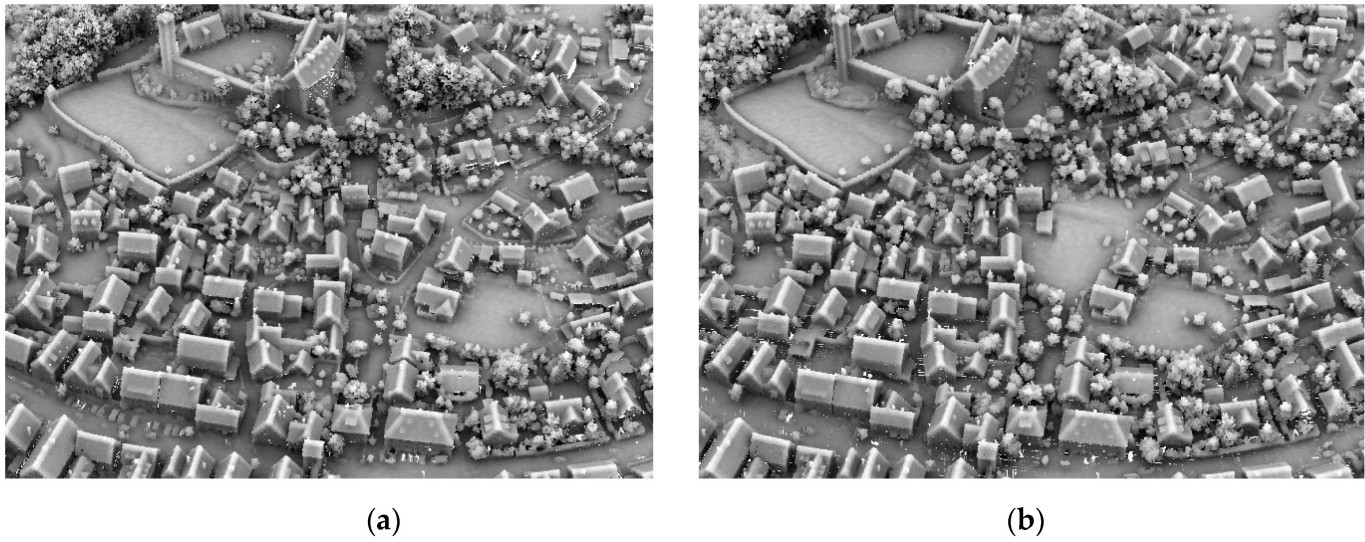

(**a**) (**b**)

**Figure 13.** (**a**) Adenberg data 2008 and (**b**) Adenberg data 2009.

4D objects by changes. This dataset contains an hourly dataset of Terrestrial Laser Scanning (TLS) point clouds acquired in the frame of coastal monitoring at the sandy beach of Kijkduin (52°04′14″ N, 4°13′10″ E), Netherlands, over a period of five months [143,144]. Link: https://doi.org/10.11588/data/4HJHAA (last access on 28 August 2022).

ICRA 2017—Change Detection Datasets. This contains three indoor datasets (living room, office, lounge) acquired using Google Tango tablets, mainly their RGB-Depth sensors with an operating range of 0.4 m to 4.0 m, and with a resolution of 320 × 180 pixels, as specified in [145]. The first one (living room, as in Figure 14) contains nine hand-held observations (9 epochs) in a controlled indoor environment. The second one (office) consists of four recordings of a controlled office environment recorded from the center of the room using a tripod. The third one consists of ten hand-held observations in an uncontrolled environment, a highly accessible meeting area observed over the course of two weeks. The link to the dataset is https://projects.asl.ethz.ch/datasets/icra2017changedetection (last accessed on 28 August 2022).

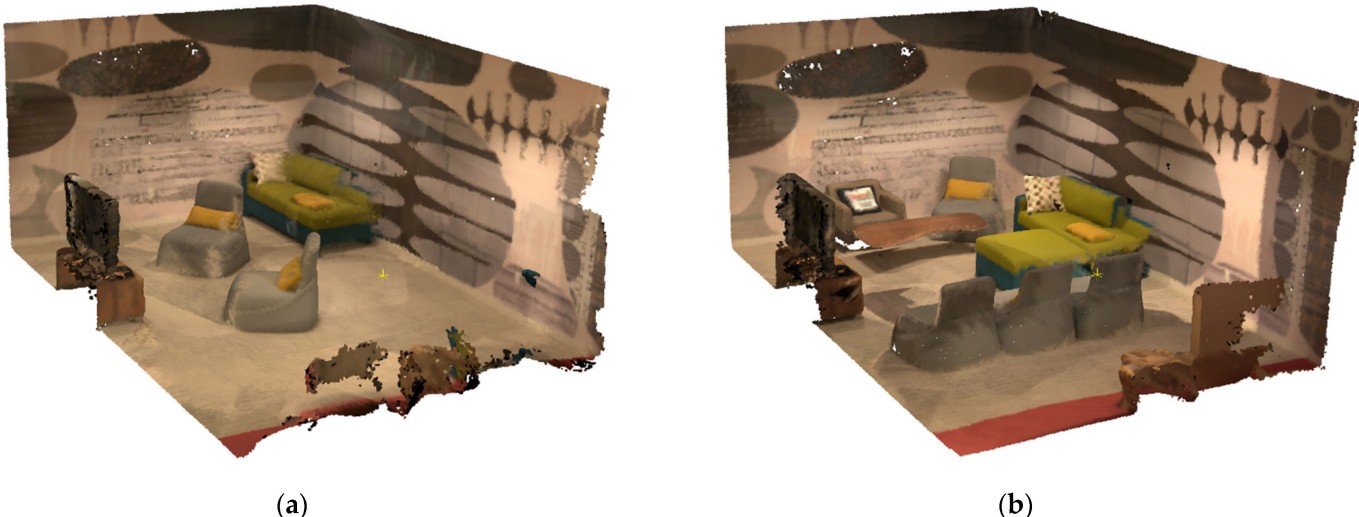

(**a**)          (**b**)

**Figure 14.** Observations at two epochs ((**a**) and (**b**)) of the living room (controlled environment).

PLS dataset of Kijkduin beach–dune. A high-resolution 4D terrestrial laser scan dataset of the Kijkduin beach–dune system, Netherlands [146]. The beach was scanned hourly for 190 days. between 11 November 2016, and 26 May 2017, by a Permanent Laser Scanner (PLS) mounted in a permanent place at 38 m height above the mean sea level. The dataset is georeferenced and contains 4082 hourly point clouds, each one containing between one and ten million 3D points. It contains additional attributes, such as the laser return and intensity.

CG-PB-M3C2. This dataset includes six points clouds acquired bi-weekly by a TLS within the summer of 2019 in the lower tongue area of the rock glacier Außeres Hochebenkar, Austria. The dataset contains approximately 222 million points per epoch, with a point spacing ≤ 0.01–0.11 m (mean 0.03 m). More details on the point clouds and their spatial coverage can be found in [147] and at https://doi.org/10.11588/data/TGSVUI (last accessed on 28 August 2022).

Near-continuous 3D time series. This dataset contains data to perform spatiotemporal segmentation in time series of surface change data for synthetic data and hourly snow cover changes acquired by a terrestrial laser scanner (TLS) [148]. The link to the dataset is https://doi.org/10.11588/data/1L11SQ (last accessed on 28 August 2022).

Change3D Benchmark. The data is provided by CycloMedia. It consists of annotated "points of interest" in street-level colored point clouds gathered using vehicle-mounted LiDAR sensors, in 2016 and 2020, in the city of Schiedam, Netherlands [121]. This dataset focuses on street furniture, with most of the labels corresponding to traffic signs (Figure 15). Although other objects, such as advertisements, statues, and garbage bins, are also included. Labeling was done through manual inspection. The dataset proposed over 78 annotated street-scene 3D point cloud pairs. Each point cloud pair represents a street scene in two different years and contains a group of changed or unchanged objects. Each object pair is assigned one of the following labels: (1) No change, (2) Added, (3) Removed, (4) Change and (5) Color change. The link to the dataset is https://kutao207.github.io/ (last accessed on 28 August 2022).

TUM City Campus—MLS test dataset. This dataset is situated in Munich, Germany (48.1493° N, 11.5685° E), and covers an area of about 29,000 m². The first one was acquired on 18 April 2016, using Mobile Laser Scanning (MLS), resulting in more than 8000 scans (rotations of the scanner head) with 1.7 billion points [149,150]. An Additional epoch (TUM-MLS-2018) was acquired on 19 December 2018, resulting in 10,500 scans (rotations of the scanner head) with 2.2 billion points. Parts of the two datasets (Figure 16) were labeled and contain semantic information (Artificial Terrain, Natural Terrain, High Vegetation, Low Vegetation, Building, Hardscape, Artifact, and Vehicle). The authors have added a less

dense, old epoch (TUM-ALS-2009), acquired by airborne laser scanning in 2009, which offers further research possibilities. So, using these multi-temporals datasets, methods for 3D CD can be developed and tested. The datasets and annotations can be downloaded at: http://s.fhg.de/mls1 (last accessed on 28 August 2022).

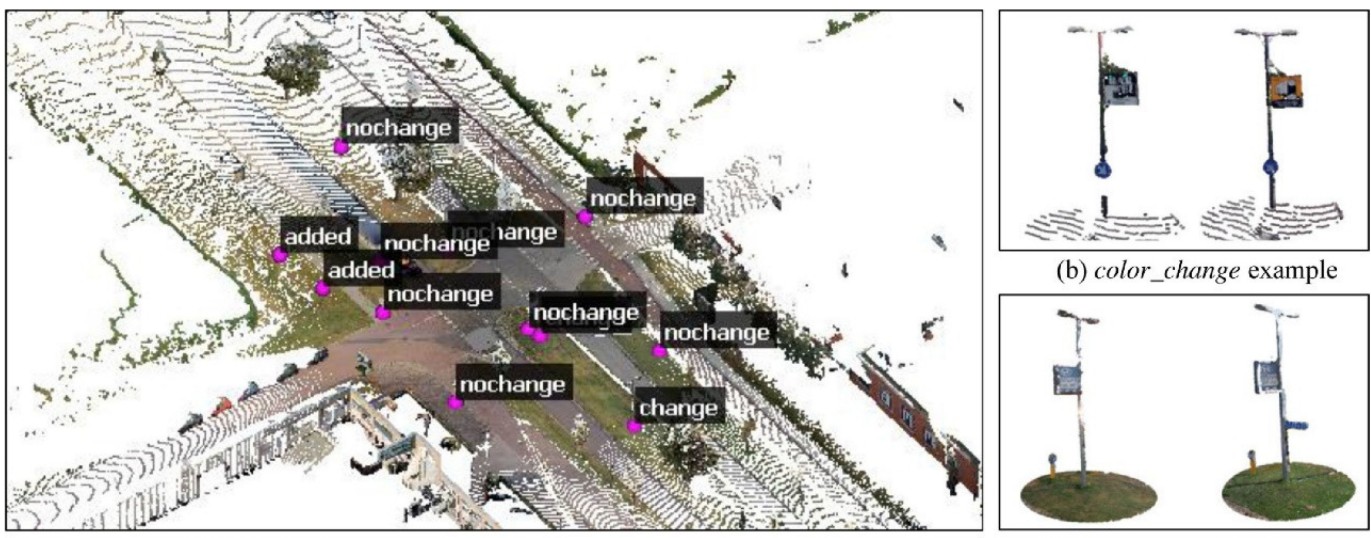

**Figure 15.** Change3D dataset with labeled points and two examples of change in color and addition. Source: Image adopted from [121].

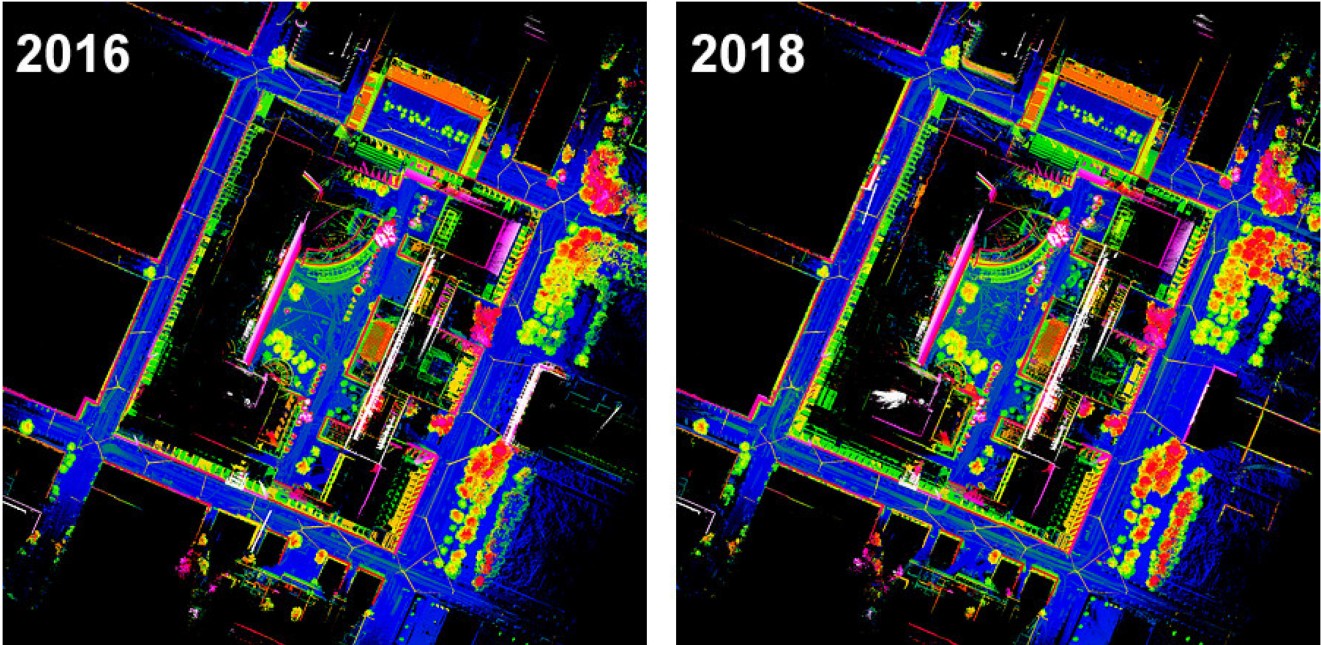

**Figure 16.** Top views of the dataset TUM-MLS-2016 and TUM-MLS-2018. Color code based on height. Source: Image adopted from [150].

URB3DCD. De Gélis et al. [151] proposed a bi-temporal simulated 3D dataset for CD in urban areas. They prepared five sub-datasets containing simulated pairs of 3D annotated point clouds with different characteristics, from high to low resolution, with various levels of noise [122].

The 2017 Change Detection Dataset. This is a dataset for visual CD consisting of images and 3D models. The dataset contains five different scenes. For each scene, it

provides a 3D model and a set of images depicting a structural element not present in the model (see Figure 17). In addition, for each scene, it provides an XML file containing the calibration of the camera used to take the pictures, as well as the extrinsic poses in world coordinates. Finally, for each image, it also provides the ground truth obtained by manually labeling the areas of change [152].

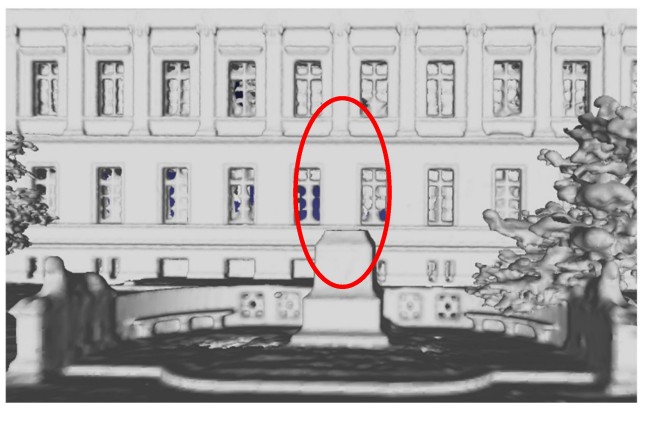
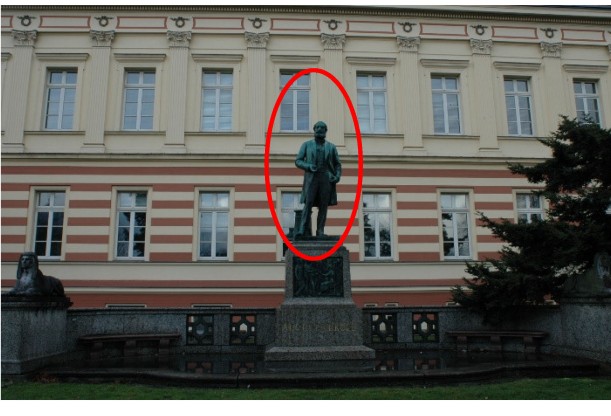

(**a**)                                                                                       (**b**)

**Figure 17.** Finding changes in the environment based on an existing 3D model (**a**), and a sequence of (currently recorded) images (**b**).

We summarize in the Table 3 the different cited datasets we found in the state of the art. For each dataset we specify if it contains a classification (class label, such as building, tree, etc.) and if it contains a ground truth about the change (change label, such as changed, unchanged, etc.). We also add, when available, the dates of acquisition and the references to the works that created these datasets.

**Table 3.** Summary of the existing datasets.

| Dataset | Class Label | Change Label | Years | Reference |
|---|:---:|:---:|:---:|:---:|
| OpenTopography | | | Multiple years | [138,139] |
| AHN1, AHN2, AHN3, AHN4 | X | | Multiple years | [141] |
| Abenberg—ALS test dataset | X | | 2008–2009 | [142] |
| 4D objects by changes | X | | 2017 | [143,144] |
| ICRA 2017—Change Detection Datasets | | | 2017 | [145] |
| PLS dataset of Kijkduin beach-dune | | | 2016–2017 | [146] |
| CG-PB-M3C2 | | | 2019 | [147] |
| Near-continuous 3D time series | | | | [148] |
| Change3D Benchmark | | X | 2016–2020 | [121] |
| TUM City Campus—MLS test dataset | X | | 2009–2016–2018 | [150] |
| URB3DCD | X | X | | [122] |
| The 2017 Change Detection Dataset | | X | 2017 | [152] |

*3.2. Evaluation Metrics*

The evaluation of the performance of a CD algorithm is a critical issue. CD algorithms process highly unbalanced data with respect to the ratio of changed to unchanged regions. Here, we introduce commonly used evaluation metrics. A well-known uncertainty metric for CD quantification is the level of detection (LODetection) [28,153–155]. It is used as a threshold to consider only real changes (where the magnitude distance is superior to the LODetection at a specific confidence interval) for further analysis and interpretation (using a statistical *t*-test and an assumption of normal distribution of errors). It is determined by the number of points ($n$) in the point set and the variance of their distances to the

fitted plane ($\sigma^2$), and by the registration error (alignment). It is defined by the following formula [130]:

$$\text{LODetection}_{\text{confidence interval}} = \pm 1.96 \times \left( \sqrt{\left( \frac{\sigma_A{}^2}{n_A} \right) + \left( \frac{\sigma_B{}^2}{n_B} \right)} + \text{alignement} \right) \quad (2)$$

where, A and B are the points clouds at epoch A and epoch B; $\sigma^2$ is the plane fitting variance (surface roughness); n is the number of points in the fitting neighborhood; alignment is the registration error, which can be estimated by using the absolute mean distance between two points clouds plus the standard deviation of these distance measurements.

Estimating the uncertainties through the process of CD is an important part of the workflow. It allows the differentiation between significant and non-significant changes. A change is considered significant when the quantified change magnitude is superior to uncertainty (LODetection). A change is considered insignificant when the magnitude of change is inferior to, or equal to, the uncertainty.

To quantitatively evaluate change detection methods, there are several metrics used in the literature [112]. Two of these metrics, commonly used to compare the results of these methods with reference data, are correctness and completeness [112]:

$$\text{Correctness} = \frac{\text{TDN}}{\text{DN}} \times 100\%$$
$$\text{Completness} = \frac{\text{TDN}}{\text{RN}} \times 100\% \quad (3)$$

where TDN (True Detected Number) is the number of real changed objects (e.g., buildings) correctly detected as changed, DN (Detected Number) is the number of changed objects (e.g., buildings) in the detected results and RN (Reference Number) is the number of changed objects (e.g., buildings) in the reference data. Since these metrics take the object as a unit of comparison, as long as the detected object has an overlap with the reference object, it is considered as correctly detected [112].

The confusion matrix is another commonly used metric for quantitative analysis of binary or multiple classification. The common definition of the confusion matrix is presented in Table 4. Where FP (false positive) and FN (false negative) refer to the points that were incorrectly classified as changed and unchanged, respectively. TP (true positive) and TN (true negative) represent the changed points and unchanged points that were correctly detected, respectively.

**Table 4.** Simple example of matrix confusion of a binary change.

| Detected | Reference | |
|---|---|---|
| | **Changed** | **Not Changed** |
| Changed | TP | FP |
| Not changed | FN | TN |

From this matrix, we can derive the most used evaluation metrics in CD, which are overall accuracy, precision, recall, F1-score, and intersection over union, as shown in Table 5. Generally, higher precision indicates fewer false prediction results, and higher recall indicates that fewer changes were missed. Furthermore, the larger their values are, the better the prediction results will be.

**Table 5.** Definitions of evaluation metrics for 3D point cloud CD.

| Metric | Description | Equation |
|---|---|---|
| Overall accuracy | It is the general evaluation metric for prediction results. | $OA = \frac{TP+TN}{TP+FP+TN+FN}$ |
| Precision | It measures the fraction of detections that were changed. | $Precision = \frac{TP}{TP+FP}$ |
| Recall | It measures the fraction of correctly detected changes. | $Recall = \frac{TP}{TP+FN}$ |
| F1 score | It refers to recall and precision together. | $F1 = \frac{2\times Precision \times Recall}{Precision+Recall}$ |
| Intersection over union | Or the Jaccard Index. | $IoU = \frac{TP}{TP+FP+FN}$ |

The first type of metrics (LODetection) is used mainly for the standard approaches, while the confusion matrix and derived metrics in Table 5 are used for learning approaches. Another important, but often neglected, metric is the calculation time, as well as the complexity of the workflow and the number of steps and operations it contains.

## 4. Discussion and Perspectives

In this review, we have subdivided 3D change detection methods into three categories. The first type includes standard methods (also called distance-based methods), the second type includes machine learning-based methods that use handcrafted features, and the last type includes the recent deep learning methods that extract more abstract features without user specification. Each method of these three categories has its advantages and limitations. To reveal the best-performing ones, in terms of metrics, a comparison is needed. I. de Gélis et al. (2021) performed a comparison between three methods: one based on distance calculation, one based on machine learning with hand-crafted features and one based on deep learning [19]. Experimental results on the URB3DCD dataset showed that the machine learning method with hand-created features using random forest gave the best results, which, nevertheless, required supervision and a feature extraction step. Tao et al. [121] conducted a benchmark of three methods, with one handcrafted feature (PoChaDeHH) and the other two learning-based (HGI-CD and SiamGCN). The results showed that the handcrafted algorithm had balanced performance over all classes. Learning-based methods achieved overwhelming performance but suffered from the class-imbalanced problem and might fail in minority classes. SiamGCN solved the class-imbalanced problem by adopting randomized oversampling and proposed a well-designed Siamese graph convolutional network architecture for the 3D CD. Comparison results showed that SiamGCN achieved the best performance on the released Change3D benchmark.

It is already well known that deep learning techniques show good performances in several point cloud processing tasks (semantic segmentation, object detection and recognition, object tracking, etc.). These techniques have demonstrated the same good performance over traditional methods in 3D CD tasks. Nevertheless, several problems are still related to this, which we summarize as follows:

Labeled data. Even though deep learning algorithms can learn highly abstract feature representations from raw 3D point clouds, successful detection and identification depend on large training samples. However, labeled high-resolution point cloud datasets are rare. Creating datasets for CD is more complicated than other tasks; it requires traversing point clouds not just at one epoch, but at two or more epochs. Given the lack of annotated data, to properly train the models, researchers use a variety of strategies, including transfer learning [156], data augmentation [157,158], and Point Clouds Generative Adversarial Networks (PC-GAN) [159,160]. Although these techniques alleviate some of the problems, coupled with a lack of samples, further improvements are still needed. Thus, future work should focus on creating methods that involve relying on small training datasets for supervised CD. This approach seems very interesting, as it minimizes the need for labeled training data.

Large-scale data. In general, change happens on a reduced area and not on the entire point clouds. The lack of prior knowledge about the precise location of change, as well as the

direction of change, means conventional unsupervised methods are unable to solve change quickly. More advanced studies are needed to solve this problem, such as hierarchical, weakly supervised, and semi-supervised methods.

Level of detail. Point clouds are massive data that can cover an entire country. These multi-temporal data collections are an ideal source for 3D change detection. The major problem is how to process such data, as change usually only occurs over limited areas. Level of detail structuration (LOD) is an efficient technique to address large data size. The idea is to use levels of detail from the highest to the lowest, so when more detail is required, the level of detail is lowered. The part that still needs to be studied is how the level of detail influences the results of the change detection. What metrics do we use to know the optimal LOD and to estimate the quality of the change accuracy?

Thus, even if the challenges related to point cloud processing are overcome, due to the availability of processing techniques, there remain many other fundamental issues related to their use in change detection that need more investigation, mainly the following:

- The use of heterogeneous and multi-modal data (acquired by photogrammetry, laser scanner or other acquisition techniques).
- The use of multi-resolution data (acquired by sensors with different specifications).
- Handling near real-time laser scanning with a high temporal resolution that has become available today. Scene flow methods can play an important role in handling this data [48–50].
- The availability of benchmark data for the 3D CD.
- The exploitation of the progress made in the 3D semantic segmentation to integrate this information in the 3D CD process.
- The use of graph neural network for change detection [15,161,162].

Finally, it is worth pointing out that 3D CD is an active research field, which aims to reach robust methods to recognize dynamics and changes in any environment using dense point clouds. This literature review shows that this is a field that requires further research to improve the performance and accuracy of the CD results. The construction of these processes from point clouds data requires designing new approaches capable of detecting changes in the earth's surface with high accuracy and efficiency. Our future research tries to respond to these challenges by proposing, as a continuation of this work, a new approach that aims to enhance the quality of CD using prior semantics in 3D Point clouds. The objective is to improve the quality of true change and its characterization.

## 5. Conclusions

The detection of change and its characterization is an essential step for monitoring dynamics on the earth's surface. In this paper, we presented a comprehensive review of change detection (CD) using 3D point clouds. We reviewed the methods used in the literature, and proposed several ways to classify them, and we highlighted the advantages and disadvantages of each category compared to the others. We proposed a first categorization based on classification and CD steps. Some start with CD and then classification, others do the reverse and others integrate the two steps at the same time to avoid errors propagation from one step to the next one. The second categorization is based on the used algorithm, Then, a third proposed classification categorizes existing methods to ones based on distance (C2C, C2M, M3C2, CD-PB-M3C2), machine learning, and deep learning.

We also revised the evaluation metrics for CD. Two categories are shown, LODetection and classification metrics derived from the confusion matrix. The first one is for methods that are based on the calculation of distances, and the others are for the evaluation of change types from learning process. With the trend toward the use of learning methods, we also reviewed and described point cloud benchmarks available in open access. We noted that deep learning methods provide more accurate results than standard methods, but at the expense of the required labeled datasets.

**Author Contributions:** Conceptualization, A.K. and R.B.; writing—original draft preparation, A.K.; writing—review and editing, A.K., R.B., Z.B., R.H., F.P.; supervision, R.B., F.P. All authors have read and agreed to the published version of the manuscript.

**Funding:** Abderrazzaq Kharroubi is an Aspirant of the Fonds de la Recherche Scientifique FNRS.

**Acknowledgments:** The authors sincerely appreciate that academic editors and reviewers give their helpful comments and constructive suggestions.

**Conflicts of Interest:** The authors declare no conflict of interest.

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
