# Peer review of "Three Dimensional Change Detection Using Point Clouds: A Review"

_2673-7418, doi:10.3390/geomatics2040025_

Round 1

Reviewer 1 Report

The paper presents a review on several approaches concerning the change detection (CD) using point clouds.
The paper is well structured in giving an overview on CD methods, available benchmark datasets, and evaluation metrics.
The paper ends with a discussion on possible future challenges.

I could imagine that the content could be of interest to beginners in this field. Although a deep understanding of
the CD methods will not be possible after reading this paper (which is not the aim of a review paper anyway) and one has to look further in the cited papers.

I stumbled about a few shortcomings (listed below), especially regarding the chosen terms, which motivated me to assign a Major Revision.

More general comments:

- The paper mentions often the term "image(s)" without being specific if the original (central perspective) image of the camera are meant or remappings (i.e. orthophotos).
  The latter would solve some of the problems associated with (original) images (e.g. view point dependences in row 172).

- There are a few English errors made in the paper: e.g. "points cloud", "data.s", "One major source ... is non-reflectance regions" ...

- Chapter 2.1.1. is titled "Data type challenges". I found this a bit vague and inappropriate. Maybe "Acquisition challenges" would be a better term.
  Additionally, 2.1.2. is termed "Processing challenges" which is also vague (and does explicitly not include "registration", which may research fields
  would include as part of "processing"). However, the four challenges listed (Irregularity, unstructured, unorderdness and rigid trafo) are characteristics
  of 3D data which are more or less independent of the purpose of the data. Thus "Data type challenges" would maybe be here a better term.

- Chapter 3.1. You say "We classify those existing datasets related to our topic into two main types: unannotated datasets ... and annotated datasets."
  However, from the description of each data set it is not very obvious to me if the annotation is provided or not. Also, a dataset table (similar to Tab. 2)
  would be good where these properties of the data sets could be presented neat and comprehensibly.

Some detailed comments:
- r73: Why is 3D scene flow not considered in this paper?

- r115: How are intra- and extra-class changes defined?

- r119: How can specularity happen on non-reflectance regions? Would "non-diffuse reflectance regions" be better?

- Chap. 2.1.2: You are referencing Fig. 2 instead of Fig. 1.

- Tab. 1: What is the difference between Z and H? What is meant with "unavailable data source"?. Typo "59sss"

- Tab. 2: The column "change detection approach" lists "post-classification" and "pre-classification". Despite these names, they are still
  ambiguous. Does pre-classification mean that the classification is done pre to CD or is the CD done pre to classif.? Some entries have
  an additional "DSM" by them. This would call for another column that differs between eg. rastered data and original point cloud.

- r220: "standard methods called distance-based methods" That's an odd wording. Pick one term. Or did you mean "standard methods (also called distance-based)"?

- r232: Chap. 2.2.1 refers Fig. 2 and mentions "DEM of differences (DoD)". But DoD is not mentioned in Fig. 2.

- Fig. 4: The subfigures c. and d. appear actually identical, although the sub-captions are different. Perhaps you could indicate the "uncertainty"?

- Fig. 6: Why is every building "removed" but the left tree is "added"?

- Fig. 9: Should one line be highlighted as "epoch A" and "epoch B"? Some added explanation to what is happening in this figure would be welcomed, especially
          explaining what the acronyms mean.

- Fig. 10. Can you add a short explanation why the starting (sub)images for "LiDAR" and "DIM" are different, but from (sub)image three onwards they are the same?

- r422: What do you mean with "performance"? Accuracy, speed, ...?

- r484: "in a permanent place at 38m." ... What? Height, distance to the beach, ...?

- r493: "teen" ?

- r515: "year 2015 and 2020" in mentioned twice.

- There is a literature reference missing for the source of Equation (1).

- The formula for F1 in Tab. 4 is obviously wrong.

- r592: "Each method of these three categories has its advantages and limitations ..."
  --> Which three categories?

- r661: "We also revised the evaluation metrics for CD methods. Two categories are shown, 661 one for methods that are based on"
  --> Which categories?

Author Response

The authors would like to thank the reviewer for the reading and the comments for improving this manuscript. Please find below the answers to each comment and the modifications made.

Point 1: The paper mentions often the term "image(s)" without being specific if the original (central perspective) image of the camera are meant or remappings (i.e. orthophotos).
The latter would solve some of the problems associated with (original) images (e.g. view point dependences in row 172).

Response 1: It is true that most of the studies we identified use ortho-photos, but there are a few that use aerial images directly (especially for older missions). So, in order to be general, we have opted for “image”. To follow the reviewer's recommendations, we have added in table 2 an indication 'ortho' to specify that it is an ortho-photo and 'original' to indicate original images.

Point 2: There are a few English errors made in the paper: e.g. "points cloud", "data.s", "One major source ... is non-reflectance regions" ...

Response 2: The necessary English corrections are made.

Point 3: Chapter 2.1.1. is titled "Data type challenges". I found this a bit vague and inappropriate. Maybe "Acquisition challenges" would be a better term.
  Additionally, 2.1.2. is termed "Processing challenges" which is also vague (and does explicitly not include "registration", which many research fields would include as part of "processing"). However, the four challenges listed (Irregularity, unstructured, unorderdness and rigid trafo) are characteristics of 3D data which are more or less independent of the purpose of the data. Thus "Data type challenges" would maybe be here a better term.

Response 3: We have made the following changes as proposed by the reviewer. Thus, " 2.1.1 Data type challenges " was replaced by " Acquisition challenges ", " Processing challenges " was replaced by "3D point clouds specifities" and “2.1 Challenges and problems” was replaced by “2.1. Challenges and specificities”.

Point 4: Chapter 3.1. You say "We classify those existing datasets related to our topic into two main types: unannotated datasets ... and annotated datasets."
  However, from the description of each data set it is not very obvious to me if the annotation is provided or not. Also, a dataset table (similar to Tab. 2) would be good where these properties of the data sets could be presented neat and comprehensibly.

Response 4: For each dataset, we have provided the proper description. Indeed, the annotation exists on two aspects. The first one is the classification into semantic classes (building, tree, ground…), and the second one is the classification of changes (changed, unchanged…). To better clarify this, we have added a table 3. that summarizes these specificities as requested by the reviewer.

Some detailed comments

r73: Why is 3D scene flow not considered in this paper?

Response: Indeed the scene flow methods are adapted to situations where movement is fast (high frequence) as in the case of autonomous vehicles or robotics. In the case of remote sensing applications, the approach is a little bit different, especially, the considered change takes place over a few days or even years. Taking into account the importance of scene flow approaches, we have mentioned the articles that deal with this issue so that the reader can investigate it further. We have also added scene flow in the perspective part, especially with the availability of high-frequency data from permanent laser scanners.

r115: How are intra- and extra-class changes defined?

Response: This is an expression error, we have corrected it to “inter and intra class” instead. In fact, the change in the acquisition conditions can generate changes in the same class so intra-class changes (for example the presence of an artifact on a building can cause the detection of a false change within the same class). This same problem can be present between different classes, hence the name of inter-class change.

r119: How can specularity happen in non-reflectance regions? Would "non-diffuse reflectance regions" be better?

Response: Yes, “non-diffuse reflection regions” is better placed especially since we are talking about specularity and absorption at the same time.

Chap. 2.1.2: You are referencing Fig. 2 instead of Fig. 1.

It was an automatic referral not updated, we have corrected it.

Tab. 1: What is the difference between Z and H? What is meant with "unavailable data source"?. Typo "59sss"

Indeed it is a notation of ellipsoidal and orthometric height, as there are other types of height ... to avoid this confusion, we have deleted (Z or H).

Open access point cloud data are still limited compared to satellite images (e.g. Landsat and sentinel). But, since more and more of these data are being made available through government programs and are open to the public, we have changed "unavailable data source" to "limited data availability".

We have corrected the typo.

Tab. 2: The column "change detection approach" lists "post-classification" and "pre-classification". Despite these names, they are still ambiguous. Does pre-classification mean that the classification is done pre to CD or is the CD done pre to classif.? Some entries have an additional "DSM" by them. This would call for another column that differs between eg. rastered data and original point cloud.

Pre-classification methods start with change detection and then classification, those called post-classification do the inverse starting with classification and then change detection while integrated methods combine the two steps at the same time to avoid the propagation of errors from one step to the other.

Exactly, there are methods that transform the point cloud into DSM before performing change detection on a raster data base. So to keep the table simple and to limit the number of entries, we have chosen to mention it this way. An explicative sentence on this was added following the reviewer's recommendations.

r220: "standard methods called distance-based methods" That's an odd wording. Pick one term. Or did you mean "standard methods (also called distance-based)"?

Yes, we mean "standard methods (also called distance-based)". We have corrected it.

r232: Chap. 2.2.1 refers Fig. 2 and mentions "DEM of differences (DoD)". But DoD is not mentioned in Fig. 2.

We have added DoD in figure 2. Indeed, the DoD (DEM of Difference) method is based on a transformation of the point cloud into a DEM (Digital Elevation Model), so it is not in the range of methods that are based on the point cloud directly but it was necessary to mention it because of its wide use.

Fig. 4: The subfigures c. and d. appear actually identical, although the sub-captions are different. Perhaps you could indicate the "uncertainty"?

Indeed the two steps are different. step c. consists in the search of corresponding planes between the point cloud A and B, while step d. consists in a distance calculation based on these planes in order to estimate the change extent. We have changed the design on c. and d. to better indicate that it is different.

Fig. 6: Why is every building "removed" but the left tree is "added"?

It was confusing because the class " removed building" can be given to either the A or B point cloud depending on which is the reference. To avoid confusion, we deleted the A point cloud and kept the recent one to give it the change and the semantic class.

Fig. 9: Should one line be highlighted as "epoch A" and "epoch B"? Some added explanation to what is happening in this figure would be welcomed, especially explaining what the acronyms mean.

Ok, we have added a paragraph that explains what the two entries take as input (point clouds at epoch A and B). We also explain acronyms' meanings.

Fig. 10. Can you add a short explanation why the starting (sub)images for "LiDAR" and "DIM" are different, but from (sub)image three onwards they are the same?

The inter-epoch features are extracted by Conjugated Ball Sampling (CBS) and concatenated to make change inferences. When the sampling centroids are determined in the LiDAR points, the samples in the DIM data are taken at the same centroids as in the LiDAR data. This ensures that the local feature vectors taken from the LiDAR data and the DIM data are always corresponding to each other. For the decoder part, the DIM feature vectors are interpolated to the raw LiDAR point location, instead of the raw DIM point locations.

We have added explicative lines on this part, with citation of the appropriate reference.

r422: What do you mean with "performance"? Accuracy, speed, ...?

By performance, we mean the evaluation metrics used to evaluate the results (accuracy, Rcall, IoU...), including time/speed as specified by the reviewer.

r484: "in a permanent place at 38m." ... What? Height, distance to the beach, ...?

At 38 m height above the mean sea level.

r493: "teen" ?

Ten (10), it was just a typo

r515: "year 2015 and 2020" in mentioned twice.

This repetition is corrected.

There is a literature reference missing for the source of Equation (1).

The reference is added.

The formula for F1 in Tab. 4 is obviously wrong.

You’re right, the equation was wrong, because of a typo a * was inserted instead of +.

It is now corrected.

r592: "Each method of these three categories has its advantages and limitations ..."
  --> Which three categories?

These are the three categories we proposed for change detection methods. Since this is a new section, we have added these phrases to reintroduce them “In this review, we have subdivided 3D change detection methods into three categories. The first type includes standard methods (also called distance-based methods), the second type includes machine learning-based methods that use handcrafted features, and the last type includes the latest deep learning methods that extract more abstract features without user specification.”

r661: "We also revised the evaluation metrics for CD methods. Two categories are shown, 661 one for methods that are based on"
  --> Which categories?

The two categories mean LODetection and the metrics derived from the confusion matrix (like precision, rcall, f1-score...), we have added an explicative sentence.

Reviewer 2 Report

This is a review manuscript on 3D change detection with a 3D point cloud as the data source. The authors claim to provide an in-depth review of research methods and research perspectives in the field, including a review of methods related to machine learning and deep learning, a summary and analysis of some recent research. In addition, some datasets of 3D point cloud change detection acquired by different sensors are listed. Finally, open research ideas and questions in this area are presented.

The article is generally well written. However, I think its structure is confusing and lacks depth and novelty, and it needs major revisions to its structure and content.

Here are my main comments and some minor ones. Each comment begins with a line number. 

1.       Regarding the structure of the article, I think further adjustments are needed to conform to a stronger logic. The current structure of the article is confusing. For example, the introduction of the dataset should be at the top, after which the preprocessing of the data and the methods should be introduced, and then the methods of change detection should be introduced after that.

2.       31: More survey reports on the application aspects of point cloud intelligence should be included. Using 3D point clouds for change detection, details about the point clouds themselves need to be described in the introduction section.

3.       49: A thorough investigation of the research background on deep learning aspects of point cloud semantic segmentation should be made. It is not in accordance with the standard of writing scientific papers to draw conclusions from only one article. Moreover, the article "Change detection within remotely sensed satellite image time series via spectral analysis." is not a review article on point cloud semantic segmentation.

4.       63: The problems facing change detection are diverse, and the summary here is one-sided. At the very least, the research points of change detection include binary change, direction or magnitude of change, "From-to" information of change, probability of change, etc. I think you should add more comments on these aspects in your introduction.

5.       101: An introduction to the pre-processing of point clouds is mandatory, and although there are many available standard datasets, an introduction to the basic experimental steps is indispensable as a review article.

6.       107: what do you mean by “data.s” here?

7.       109: This section summarizes the factors influencing the use of point clouds for change detection in three ways, but such a simple description has very little scientific value. I suggest a table to summarize all the information more clearly and accurately. This summarized information can be much richer, it can include the sources of different influencing factors, the problems generated, and the solutions corresponding to different studies, etc.

8.       140: The authors here summarize four challenges regarding data processing, but do not describe some of the more representative research methods. That is, which research method addresses which of the above challenges. Since these kinds of challenges exist, is there a corresponding solution? If so, then the authors should list them.

9.       198: Four ways to classify change detection techniques are listed here, but the following summarizes the literature researched based on three criteria: input data, classification and order of change detection, and target. This is very illogical. How did you use the two criteria of change unit and technology use?

10.    224: Among the deep learning methods listed are network models based on convolutional neural networks. However, as of now, graph neural network models are also used in the study of point cloud change detection. Other non-convolutional types of network models need to be added here.

11.    226ff: Each method has its limitations and it is recommended to summarize them in a table.

12.    226ff:All the listed methods have their core algorithms, and a description of the core algorithm for each article is necessary. It is suggested that how to add the description of each method's core algorithm in the paper.

13.    226ff: The authors' descriptions of each method are abbreviated and do not allow a clear understanding of the specific steps to implement each method. Also, the general steps of change detection using point clouds are not equipped with relevant schematics.

14.    436ff: Is there no "ground truth" in every dataset? If so, it should be drawn in the figure.

15.    591: The authors in the discussion section only summarize the relevant issues and do not sufficiently summarize the advantages, limitations, challenges and opportunities of the existing methods, and should give higher level insights based on the existing methods and point out the guiding significance of this work for the future development of the related fields.

Author Response

The authors would like to thank the reviewer for the reading and the comments for improving this manuscript. Please find below the answers to each comment and the modifications made.

Point 1. Regarding the structure of the article, I think further adjustments are needed to conform to a stronger logic. The current structure of the article is confusing. For example, the introduction of the dataset should be at the top, after which the preprocessing of the data and the methods should be introduced, and then the methods of change detection should be introduced after that.

Response 1: Following the reviewer's recommendations, we have added a section on preprocessing (2.2). Thus, the structure of the paper becomes as follows: we start with the specificities and challenges related to the use of point clouds and compare 3D with 2D change detection. Afterward, some preprocessing methods are detailed. Then the change detection methods are presented with a discussion of the advantages and disadvantages of each category. Finally, a list of open access datasets is proposed as well as evaluation metrics. We have kept the datasets at the end of this review, to already introduce their importance and the motivation to have more of them available for the task of change detection based on learning methods in particular. To give a logical and fluid flow between sections, we have added transition sentences.

Point 2. 31: More survey reports on the application aspects of point cloud intelligence should be included. Using 3D point clouds for change detection, details about the point clouds themselves need to be described in the introduction section.

Response 2: We have added details about 3D point clouds, in the introduction, as requested by the reviewer.

Point 3. 49: A thorough investigation of the research background on deep learning aspects of point cloud semantic segmentation should be made. It is not in accordance with the standard of writing scientific papers to draw conclusions from only one article. Moreover, the article "Change detection within remotely sensed satellite image time series via spectral analysis." is not a review article on point cloud semantic segmentation.

Response 3: This is a typo, the reference 8 should not exist in this place. We have added the right references and details about the semantic segmentation of 3D point clouds since the latter is not the core of this review. as requested by the reviewer.

Point 4. 63: The problems facing change detection are diverse, and the summary here is one-sided. At the very least, the research points of change detection include binary change, direction or magnitude of change, "From-to" information of change, probability of change, etc. I think you should add more comments on these aspects in your introduction.

Response 4: We have made the corrections as requested by the reviewer. But in this review, we were limited to the detection change “From to” without mentioning other aspects in detail.

Point 5. 101: An introduction to the pre-processing of point clouds is mandatory, and although there are many available standard datasets, an introduction to the basic experimental steps is indispensable as a review article.

Response 5: We have added a section that details the possible preprocessing of point clouds in the change detection process. Please see section 2.2

Point 6. 107: what do you mean by “data.s” here?

Response 6: It's a typo, the necessary English corrections are made.

Point 7. 109: This section summarizes the factors influencing the use of point clouds for change detection in three ways, but such a simple description has very little scientific value. I suggest a table to summarize all the information more clearly and accurately. This summarized information can be much richer, it can include the sources of different influencing factors, the problems generated, and the solutions corresponding to different studies, etc.

Response 7: We added details about this section as requested by the reviewer. Please refer to 2.1.1.

Point 8. 140: The authors here summarize four challenges regarding data processing, but do not describe some of the more representative research methods. That is, which research method addresses which of the above challenges? Since these kinds of challenges exist, is there a corresponding solution? If so, then the authors should list them.

Response 8: You are right, following this recommendation, we have added the research with references that propose solutions to the challenges described. Please see in the paper section 2.1.2.

Point 9. 198: Four ways to classify change detection techniques are listed here, but the following summarizes the literature researched based on three criteria: input data, classification and order of change detection, and target. This is very illogical. How did you use the two criteria of change unit and technology use?

Response 9: As mentioned in the text, change detection can have several basic units. There are methods that are based on the point, others on the transformation of the point into voxel, and others work with ray tracing. All the methods we have mentioned in our reviews are based on point clouds. For the technology used criteria, we have detailed it in figure 2. and the sections that follow it.

Point 10. 224: Among the deep learning methods listed are network models based on convolutional neural networks. However, as of now, graph neural network models are also used in the study of point cloud change detection. Other non-convolutional types of network models need to be added here.

Response 10: The SimaGCN we presented in section 2.3.3 is Siamese Graph Convolutional Network (SiamGCN) for 3D point cloud CD. According to literature search, graph-based methods for change detection are not yet commonly used on point clouds, unlike images. So, we have added this point, also, in the perspectives.

Point 11. 226ff: Each method has its limitations and it is recommended to summarize them in a table.

Response 11: At the end of each section, we have a summary of the limitations and advantages of each method.

Point 12. 226ff: All the listed methods have their core algorithms, and a description of the core algorithm for each article is necessary. It is suggested how to add the description of each method's core algorithm in the paper.

Response 12: In order to keep the review as efficient as possible, we have added a general description of each algorithm with a schema to illustrate how it works. Although a deep understanding of CD methods is not possible after reading this article (which is not the purpose of a review article anyway), one should look further into the cited articles.

Point 13. 226ff: The authors' descriptions of each method are abbreviated and do not allow a clear understanding of the specific steps to implement each method. Also, the general steps of change detection using point clouds are not equipped with relevant schematics.

Response 13: In order to keep the review as efficient as possible, we have added a general description of each algorithm with a schema to illustrate how it works. Although a deep understanding of CD methods is not possible after reading this article (which is not the purpose of a review article anyway), one should look further into the cited articles.

Point 14. 436ff: Is there no "ground truth" in every dataset? If so, it should be drawn in the figure.

Response 14: Not all the datasets we have presented have a ground truth for the change labels, so we have specified the annotated datasets and the non-annotated ones. For each dataset, we have provided the proper description. Indeed, the annotation exists in two aspects. The first one is the classification into semantic classes (building, tree, ground…), and the second one is the classification of changes (changed, unchanged…). For more details on this point, we have added table 3, which summarizes these specificities as requested by the reviewer.

Point 15. 591: The authors in the discussion section only summarize the relevant issues and do not sufficiently summarize the advantages, limitations, challenges and opportunities of the existing methods, and should give higher level insights based on the existing methods and point out the guiding significance of this work for the future development of the related fields.

Response 15: We have detailed the advantages, limitations, and challenges of the methods at the end of each section 2.3.1, 2.3.2, and 2.3.2. At the reviewer's recommendation, we have added a paragraph in part 4. Discussion and perspectives.

Reviewer 3 Report

The paper is well structured and covers many years of publications. 

Although this paper is a review and so it has not the purpose to present innovative methodologies or similar, I suggest a more focused discussion on all the data gathered from the survey.

Author Response

Following the reviewer's recommendation, the authors have revised the discussion section to give more focus to this part which is of great importance for a review. Thus, we have added some sections about perspectives and the discussion of the articles and data collected. We have also added parts and improved the existing ones following the recommendations of other reviewers. We hope that the article in its current form meets your expectations. Thank you again for all your efforts. 

Round 2

Reviewer 1 Report

Dear authors,

thanks for considering my comments to the 1st submission!

Author Response

The authors would like to thank the reviewer for all the pertinent remarks and efforts made to improve the quality of this review. We have checked all the remarks and we hope they will be approved.

Reviewer 2 Report

The paper has been considerably improved since the first version:

  Large parts of the paper have been improved and rewritten.

 The structure of the paper is much more logical.

 It can be accepted with minor revisions.

However, some weaknesses remain:

1.     In order to better inform the reader about nDSM, you need to give a schematic diagram for illustration.

2.     All diagrams in the article should have a uniform format. For example, you can use (a), (b)... to represent the individual subgraphs. Instead of using "left" and "right".

3.     The vast majority of the methods in Table 3 are work to detect buildings. So, what about the relevant studies of other land types? Is it possible to add some descriptions of studies of other land types? For example, forests.

4.     "Level of detail:"? Please check the punctuation throughout the text.

Author Response

The authors would like to thank the reviewer for the reading and the comments for improving this manuscript. Please find below the answers to each comment and the modifications made.

Point 1. In order to better inform the reader about nDSM, you need to give a schematic diagram for illustration.

Response 1. Following the reviewer's recommendation, we have added a schematic diagram to clarify the difference between DEM, DSM, and nDSM (please see figure 2).

Point 2. All diagrams in the article should have a uniform format. For example, you can use (a), (b)... to represent the individual subgraphs. Instead of using "left" and "right".

Response 2. We have standardized all subfigures in the text with (a), (b)... instead of (left) and (right).

Point 3. The vast majority of the methods in Table 3 are work to detect buildings. So, what about the relevant studies of other land types? Is it possible to add some descriptions of studies of other land types? For example, forests.

Response 3. Indeed, other datasets are available on topics such as trees, and forests but are based on transforming the point cloud into a digital surface model (DSM) and canopy height model (CHM). Following this recommendation, we have added 4 studies on this subject in table 2. It is worth stating that the AHN1, AHN2, AHN3, and AHN4 for example that we have mentioned, include the vegetation class. The cited study (Fekete and Cserep, 2021) used AHN2 and AHN3 for tree segmentation and change detection in large urban areas.

Fekete, A., Cserep, M., 2021. Tree segmentation and change detection of large urban areas based on airborne LiDAR. Comput. Geosci. 156, 104900. https://doi.org/10.1016/j.cageo.2021.104900

Point 4. "Level of detail:"? Please check the punctuation throughout the text.

Response 4. We corrected it. For this final version, we reviewed the entire text and made all the necessary corrections.

Thank you very much